# Unifying non-Markovian dynamics and agent heterogeneity in scalable stochastic networks

Aurélien Pélissier [1,2,3], Miroslav Phan [1,2], Didier Le Bail[4], Niko Beerenwinkel [2] & María Rodríguez Martínez [1,5] ✉

Stochastic processes underpin dynamics across biology, physics, epidemiology, and finance, yet accurately simulating them remains a major challenge. Classical approaches such as the Gillespie algorithm are exact for Markovian, time-independent systems, where propensities depend only on the current state and agents of a given type are statistically identical. While efficient, this framework misses a defining feature of many real systems: heterogeneity and memory at the level of individual agents. Cells may divide or differentiate on distinct intrinsic timescales, individuals may preferentially interact with specific partners, and inter-event-time distributions can deviate strongly from the exponential. We introduce MOSAIC (Modeling of Stochastic Agents with Individual Complexity), a general and scalable framework that embeds agent-specific properties directly into the dynamics. MOSAIC unifies heterogeneous rates, dynamic interaction preferences, and both Markovian and non-Markovian waiting-time distributions within a single stochastic formalism, while retaining Gillespie-like computational cost. Applications to delayed biochemical reactions, competitive immune-cell dynamics, and temporal social networks show that MOSAIC reproduces empirical features that existing methods either miss or capture only at prohibitive computational cost, establishing it as a practical tool for simulating heterogeneous stochastic systems.

Stochastic processes provide a powerful framework for describing the random dynamics of complex systems across biology, physics, chemistry, epidemiology, and the social sciences[1–6]. Classical approaches typically rely on the Markovian assumption, where the probability of future events depends only on the present state of the system. This assumption has enabled elegant mathematical analysis and efficient algorithms, most notably the Gillespie stochastic simulation algorithm (SSA)[7,8], which generates statistically exact trajectories for continuous-time Markov processes whose reaction propensities depend only on

the current state. It has become a cornerstone of stochastic modeling[9–11], as well as hybrid solvers such as COPASI[12] that combine deterministic and stochastic descriptions within this framework.

Yet, the simplicity of the Markovian paradigm comes at a cost. Real systems rarely behave in a memoryless way, and most cannot be reduced to propensities that depend only on the instantaneous state. Molecular, cellular, and social processes often exhibit memory and temporal correlations: molecules may undergo delays before reacting, neurons exhibit refractory periods between firings, and human activity

[1]IBM Research Europe, Rüschlikon, Switzerland. [2]Department of Biosystems Science and Engineering, ETH Zurich, Basel, Switzerland. [3]Institute of Computational Life Sciences, Zürich University of Applied Sciences (ZHAW), Wädenswil, Switzerland. [4]Centre de Physique Théorique (CPT), Aix-Marseille University, CNRS, Marseille, France. [5]Department of Biomedical Informatics & Data Science, Yale School of Medicine, New Haven, CT, USA. ✉e-mail: maria.rodriguezmartinez@yale.edu

often follows bursty patterns separated by extended intervals of inactivity[13–16]. Importantly, they also exhibit heterogeneity at the level of individual agents. Cells within the same population may divide or differentiate on different timescales, individuals may favor specific interaction partners, and inter-event time distributions (IEDs) may vary widely, from long-tailed to narrow or near normally distributed. While analytical approaches can capture certain classes of such non-Markovian effects, especially at steady state[17], simulating large, heterogeneous systems with general non-Markovian timing remains challenging. Ignoring this diversity systematically leads to simulations that deviate from empirical observations, while incorporating it markedly improves agreement with experimental and population-level data[18–20].

Several approaches have sought to overcome the limitations of simple Markovian models. Hidden Markov Models, for example, capture long-range dependencies through latent states and have been applied in domains such as climate and finance[21–23], but they demand heavy parameterization, computationally intensive inference, and often lack a clear mechanistic link to the underlying process. Other strategies modify stochastic simulation directly. Delay-based algorithms and non-Markovian Gillespie variants incorporate memory by allowing rates to depend on elapsed time or mixtures of exponentials[24–26], while network-based constructions generate bursty temporal patterns[27,28]. These methods extend the Gillespie framework to non-exponential dynamics, but typically assume homogeneous agents and incur higher computational costs, scaling as $O(\log N)$ or worse per event due to reaction-queue management or complex rate sampling.

Agent-based models (ABMs) provide a complementary path, enabling explicit heterogeneity in rules and interaction patterns[29,30]. High-performance epidemic simulators such as GEMFsim, EpiFast, EpiSimdemics, and EpiHiper likewise support agent-level heterogeneity at large scale[31–34], but typically rely on discrete-time updates, large event queues, and distributed-memory HPC resources. A notable subclass of ABMs, activity-driven network models, highlights the importance of heterogeneous activity rates[35], but generally lacks rigorous stochastic guarantees and is not optimized for efficient, exact simulation. Collectively, these approaches underscore the value of heterogeneity (and, in some cases, memory), yet there is still no general framework that combines agent-specific dynamics and non-Markovian timing in continuous time while retaining Gillespie-like simplicity and single-node scalability.

Here we introduce MOSAIC (Modeling of Stochastic Agents with Individual Complexity), a rejection-based simulation framework that preserves the efficiency of Gillespie while allowing each agent to follow its own dynamics. Importantly, MOSAIC considers all instantaneous reaction rates at the start of each iteration, which permits dynamic, state-dependent changes in interaction preferences or individual inter-event distributions. In doing so, MOSAIC addresses both major sources of complexity, memory effects that generate non-exponential waiting times and dynamic agent-level heterogeneity in rates, preferences, and inter-event distributions, while maintaining update steps that scale independently of system size. This represents a paradigm shift in stochastic modeling, enabling scalable simulations that link microscopic diversity and memory to emergent macroscopic behavior.

We demonstrate MOSAIC's capabilities through three representative applications: clonal B-cell dynamics during immune responses, RNA transcription with feedback regulation, and temporal social networks with heterogeneous node activity. Across these domains, MOSAIC reproduces empirical features that existing methods either fail to capture or capture only at a heavy computational cost, establishing it as a general and scalable framework for the next generation of stochastic simulations.

## Results

### Modeling of stochastic agents with individual complexity (MOSAIC)

At its core, MOSAIC generalizes the Gillespie algorithm by representing each agent as its own renewal process with a time-dependent reaction rate[25]. Unlike the standard Gillespie (SG) algorithm, which collapses identical particles into reaction channels with fixed exponential waiting times, MOSAIC allows every process to follow a distinct IED or to have a rate that depends on individual properties and interactions. This makes it possible to model systems where both memory effects and agent heterogeneity drive the dynamics. We denote by $N$ the total number of elementary stochastic processes (potential events) in the system. In a multi-agent model, each admissible agent–agent interaction defines one such process. Thus, $N$ is fixed by the model structure and is not a tunable parameter. A central challenge is scalability. In many realistic systems $N$ is very large, either because every individual agent must be tracked separately or because interactions are pairwise, so that $N$ grows combinatorially with the population sizes (e.g., $\mathcal{O}(N_A N_B)$ for bimolecular reactions). Treating each process as an independent channel, as in SG, would lead to prohibitive computational costs[10].

MOSAIC overcomes this barrier by tracking only a single global maximum rate, $\lambda_{\max}$, rather than updating all individual rates at every step. Candidate processes are drawn uniformly at random from the population, each with probability $1/N$, and then accepted with probability proportional to their instantaneous rate, $\lambda_j(t_j)$ (Eq. (4)). Here $\lambda_j(t_j)$ is shorthand for a state-dependent rate, which may depend both on the elapsed time $t_j$ since the last event of process $j$ and on the current configuration of the system. This rejection sampling step ensures that the effective probability of firing is correctly weighted by the true internal rates of the processes, while avoiding the need to explicitly maintain and update all rates[36,37]. By relying on a global bound rather than explicit enumeration, each update requires constant-time operations (Methods 3.1), where the complexity per accepted event depends on the expected number of rejections, $r$, which we show later is constant and independent of $N$, unlike the $O(N)$ scaling of the standard Gillespie algorithm. Moreover, MOSAIC does not require maintaining a reaction queue, further reducing memory usage and implementation complexity. This design ensures that MOSAIC remains efficient even in populations with millions of heterogeneous agents or densely interacting networks, while still capturing the full diversity of waiting times and agent-specific interaction preferences.

Algorithmically, MOSAIC explicitly tracks a global time $T$ and, for each process $j$, the time $\tau_j$ of its last event. The elapsed time of process $j$ is then given implicitly by $t_j = T - \tau_j$. Let $\lambda_j(t_j)$ denote its time-dependent (and state-dependent) rate. Each iteration of MOSAIC proceeds in four steps:

(i) *Set the global maximum rate.* Choose $\lambda_{\max}$ such that

$$\lambda_{\max} \geq \max_{1 \leq j \leq N} \lambda_j(t_j). \tag{1}$$

In practice, $\lambda_{\max}$ is a user-chosen upper bound: it can be set to the true maximum rate in the system, or increased beyond this value to improve accuracy at the cost of additional rejections.

(ii) *Advance the time.* Draw $u \sim \mathcal{U}[0, 1]$, set $\Delta t$ and advance the global time

$$\Delta t = \frac{\ln(1/u)}{N \cdot \lambda_{\max}}. \quad \text{and} \quad T \leftarrow T + \Delta t \tag{2}$$

(iii) *Select a candidate process.* Choose $j$ uniformly at random:

$$p_j = \frac{1}{N}. \tag{3}$$

(iv) *Accept or reject the event.* Compute the updated elapsed time of the candidate process as $t_j = T - \tau_j$ and accept with probability

$$p_{\text{accept}} = \frac{\lambda_j(t_j)}{\lambda_{\max}}, \qquad (4)$$

in which case the last-event time of the firing process is reset, $\tau_j \leftarrow T$. This state update may in principle change the rates of other processes, but MOSAIC does not require maintaining or updating the full vector $(\lambda_1, \ldots, \lambda_N)$; only the global bound $\lambda_{\max}$ must remain an upper bound on all instantaneous rates, which can be enforced in $O(1)$ time as described in Methods 3.1. If the event is rejected, the system state and all $\tau_k$ remain unchanged, and the only effect of the iteration is that the global time $T$ has advanced by $\Delta t$.

MOSAIC recovers the Gillespie renewal theory, in the limit $\Delta t \to 0$ (equivalently when $N \to \infty$ or $\lambda_{\max} \to \infty$), the simulated IED for process $j$ converges to the renewal-theory form:

$$\psi_j(t_j) = \lambda_j(t_j) \exp\left(-\int_0^{t_j} \lambda_j(\tau)\,d\tau\right). \qquad (5)$$

A full derivation is provided in Supplementary Section B. Although rejection introduces an approximation at finite step size, MOSAIC becomes exact in the small-$\Delta t$ limit and can reach arbitrarily high accuracy by increasing $\lambda_{\max}$, even when $N$ is modest.

In terms of accuracy, the expected error in the sampled inter-event times of a process satisfies (Supplementary Section B.3):

$$\mathbb{E}\left[\text{error}_{\text{MOSAIC}}\right] \sim \frac{\langle\lambda'\rangle}{N \cdot \lambda_{\max}^2}, \qquad (6)$$

with $\langle\lambda'\rangle = \int_0^\infty \lambda'(t)\psi(t)\,dt$ the mean derivative rate. In practice, this error correlates to the Earth Mover's Distance[38] between the true and simulated distribution of inter-event times for that process. Here $N$ is fixed by the model, whereas $\lambda_{\max}$ is freely chosen and provides a direct accuracy-efficiency trade-off.

The computational complexity is affected by the number of rejected reactions, which scales as $r \propto \lambda_{\max}/\lambda_0$ per accepted event, where $\lambda_0$ is the mean event rate (Supplementary Section B.5). Thus, increasing $\lambda_{\max}$ reduces the error quadratically, but raises the number of rejections only linearly.

Finally, MOSAIC generalizes the standard Gillespie algorithm by embedding heterogeneity directly at the level of individual agents. Each process can follow its own inter-event distribution, with rates defined as $\lambda_j(t) = \psi_j(t)/\Psi_j(t)$. This formulation allows Gamma, Weibull, Pareto, log-normal, and other biologically relevant waiting-time distributions to be simulated without approximation, enabling accurate representation of memory effects (Supplementary Section A). In addition, reaction rates can depend on agent-specific traits $\mathcal{P}_j$ and on the current system state, such as susceptibility in epidemics, binding affinity in biochemistry, or sociability in networks. Unlike traditional approaches that require discretizing these properties into subchannels[10], MOSAIC evaluates $\lambda_j = \lambda_j(t_j, \mathcal{P}_j, \text{state})$ directly, thereby preserving the full variability of individual properties.

In summary, MOSAIC extends the Gillespie framework to non-Markovian systems with full agent-level heterogeneity. By tracking only a global maximum rate, it scales efficiently to large populations and multi-reactant interactions, while capturing diverse waiting-time distributions and individual-specific properties. This combination of accuracy and scalability establishes MOSAIC as a general tool for stochastic simulation across biology, physics, and networked systems.

## Application 1: competitive selection in heterogeneous B-cell populations

B-cell affinity maturation provides a canonical example of a heterogeneous stochastic process shaped by competition. During an immune response, B cells expressing diverse antigen receptors compete for survival signals delivered by a limited pool of T cells. Clones, i.e. groups of B cells descended from the same ancestor, with higher antigen affinity are preferentially rescued from apoptosis, undergo further rounds of proliferation, and acquire affinity-enhancing mutations, ultimately giving rise to dominant clonal families (Fig. 1A). Although many cellular processes, such as apoptosis or division, are governed by intrinsic programs with characteristic timescales that deviate from exponential waiting-time distributions, we focus here on a simplified *Markovian* setting in which all reactions are assumed exponential. This approximation, which has previously been shown to capture the essential features of germinal center dynamics[10,11], enables us to isolate the role of clonal heterogeneity and to demonstrate the efficiency gains of MOSAIC over existing Markovian methods. Extensions to non-exponential dynamics will be addressed later in this article (Sections 1.3 & 1.4).

We consider a minimal model consisting of five processes (Methods 3.2): (i) binding of B cells to T cells, (ii) displacement of bound B cells by higher-affinity competitors, (iii) apoptosis of B cells that did not receive survival signals from T cells, (iv) spontaneous unbinding after sufficient T-cell signaling has been received, and (v) subsequent division with affinity updates. Among these, competition (process ii), in which B cells with higher-affinity receptors preferentially displace lower-affinity counterparts from T-cell interactions, plays the central role in affinity maturation. We represent this process as

$$[B_1 T] + B_2 \xrightarrow{(IP)} [B_2 T] + B_1, \qquad (7)$$

where *(IP)* indicates that the rate depends on the individual properties of the cells, in this case, receptor affinity. In our model, a bound B cell $B_1$ can only be displaced if the incoming competitor $B_2$ has strictly higher affinity. This pairwise competition drives clonal selection but also introduces algorithmic challenges, as each event can alter the rates of many other competing pairs. MOSAIC overcomes this challenge by using a rejection-based scheme, in which all potential interactions are sampled, but only those that satisfy the strict affinity threshold defined by the existing interaction are accepted (Eq. (4)). In this application the dynamics are purely Markovian, so the simulation is exact and we simply set $\lambda_{\max} = \max_{1 \le j \le N} \lambda_j$, i.e., equal to the true maximum rate over all processes.

Simulations with $N_B = 1000$ B cells and $N_T = 10$ T cells, modeling 50 days of affinity maturation, reproduce experimental observations. Figure 1B shows the trajectories of the ten most expanded clones in a representative run, where one clonal family (clone 2) becomes dominant after several days. When results are aggregated across replicates (Fig. 1C), the dominance of the leading clone increases monotonically, in agreement with experimental data[39].

Next, we examine computational scaling. Conventional Gillespie methods[7,8] and its variant like COPASI[12] face a combinatorial burden due to pairwise competition. After each event, it is necessary to check whether any unbound B cell has higher affinity than those currently bound to T cells, which requires maintaining and updating a list of competing pairs. Because each displacement event can change the rates of many potential competitors, this bookkeeping becomes computationally expensive. In a standard Gillespie-style implementation for this heterogeneous setting, one must treat each admissible B-T pair as a separate stochastic process with its own propensity, leading to $N = N_B \cdot N_T$ such processes and $O(N)$ complexity per step. Tree-based Gillespie[40] and DelaySSA[41] improve event selection to $O(\log N)$ through hierarchical data structures, but still require updating $k$ propensities

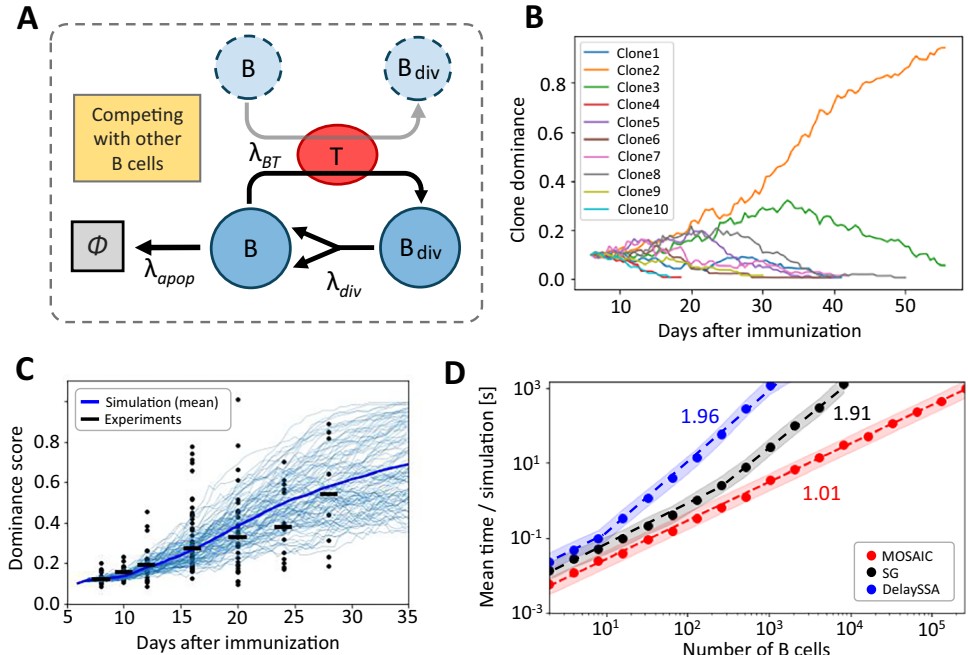

**Fig. 1 | Efficient simulation of affinity maturation with MOSAIC. A** Cartoon of affinity maturation. B cells compete for T-cell help: higher-affinity clones are rescued from apoptosis and expand, whereas lower-affinity clones undergo apoptosis. **B** Evolution of 10 clonal families in a representative simulation. Family 2 dominates after several days. **C** Dominance of the largest clone across 1000 simulations (thin blue lines = replicates, thick blue = mean). Experimental data from ref. [39] are shown as scatter points with median bars. **D** Computational performance of different algorithms as $N_B$ increases, with slopes obtained from piecewise polynomial fits. Shaded regions denote standard deviations.

**Table 1 | Comparison of stochastic simulation algorithms for systems with heterogeneous agents**

| Algorithm | Arbitrary inter-event time distribution | Dynamic delays | Accuracy | Complexity/ simulation | Complexity / time step |
|---|---|---|---|---|---|
| Standard Gillespie | ✗ | ✓ | Exact | $O(N^2)$ | $O(N)$ |
| Tree-based Gillespie[40] | ✗ | ✓ | Exact | $O(Nk \log N)$ | $O(k \log N)$ |
| DelaySSA[41] | ✓ | ✗ | Exact | $O(Nk \log N)$ | $O(k \log N)$ |
| nMGA[25] | ✓ | ✓ | Only when $N \to \infty$ | $O(N^2)$ | $O(N)$ |
| MOSAIC (ours) | ✓ | ✓ | Arbitrary high | $O(rN)$ | $O(1)$ |

$N$ is the number of processes, $k$ is the number of propensity updates per iteration, $r$ is the rejection/acceptance ratio (Supplementary Section B.5). Details on complexity derivations are provided in the Supplementary section H.

after each event, leading to an overall cost of $O(k \log N)$ that grows superlinearly with system size. MOSAIC avoids this overhead entirely by relying on the rejection scheme, where all potential interactions are sampled but only those satisfying the affinity condition are accepted (Eq. (4)). This reduces the per-iteration cost to constant time on average, independent of $N$ or $k$. Empirical benchmarks confirmed this scaling behavior (Fig. 1D; Table 1), with MOSAIC maintaining near-linear runtime even in large, densely interacting populations.

These results demonstrate that even under purely Markovian assumptions, MOSAIC faithfully reproduces the qualitative and quantitative features of affinity maturation while maintaining computational efficiency at scales where conventional methods become prohibitive. This establishes its utility for modeling heterogeneous, competitive systems.

**Application II: reactions with state-dependent non-exponential delays**

Many biological processes display memory effects, where reaction times deviate from simple exponential behavior and instead reflect intrinsic programs or multi-step mechanisms. For instance, apoptosis and cell division unfold on characteristic timescales, while transcription elongation can pause intermittently, giving rise to

broad inter-event time distributions. Classical extensions of Gillespie, such as DelaySSA[24,41], can simulate arbitrary IEDs by scheduling delays at the moment of initiation. However, once a delay has begun, it cannot be modified if system conditions change. This limitation prevents DelaySSA from capturing state-dependent waiting times, where effective delays vary dynamically with the molecular context. In contrast, MOSAIC continuously evaluates instantaneous rates, allowing each process to adapt in real time to changes in the system. This flexibility enables the modeling of feedback-regulated processes in which delays are coupled to evolving system states.

We illustrate this capability with a classic model of the transcription factor Hes1, which represses its own transcription through a delayed negative feedback loop[42–44] (Methods 3.3). In this system (Fig. 2A), RNA polymerase initiates transcription, the nascent RNA undergoes a non-exponential elongation phase before maturing into mRNA, and the mRNA is then translated into protein. The resulting Hes1 protein feeds back to inhibit further initiation, producing oscillations in RNA and protein levels.

The elongation step provides a critical test case. Experiments indicate that elongation times are broadly distributed, with heavy tails arising from polymerase pausing[45,46]. We model elongation as a

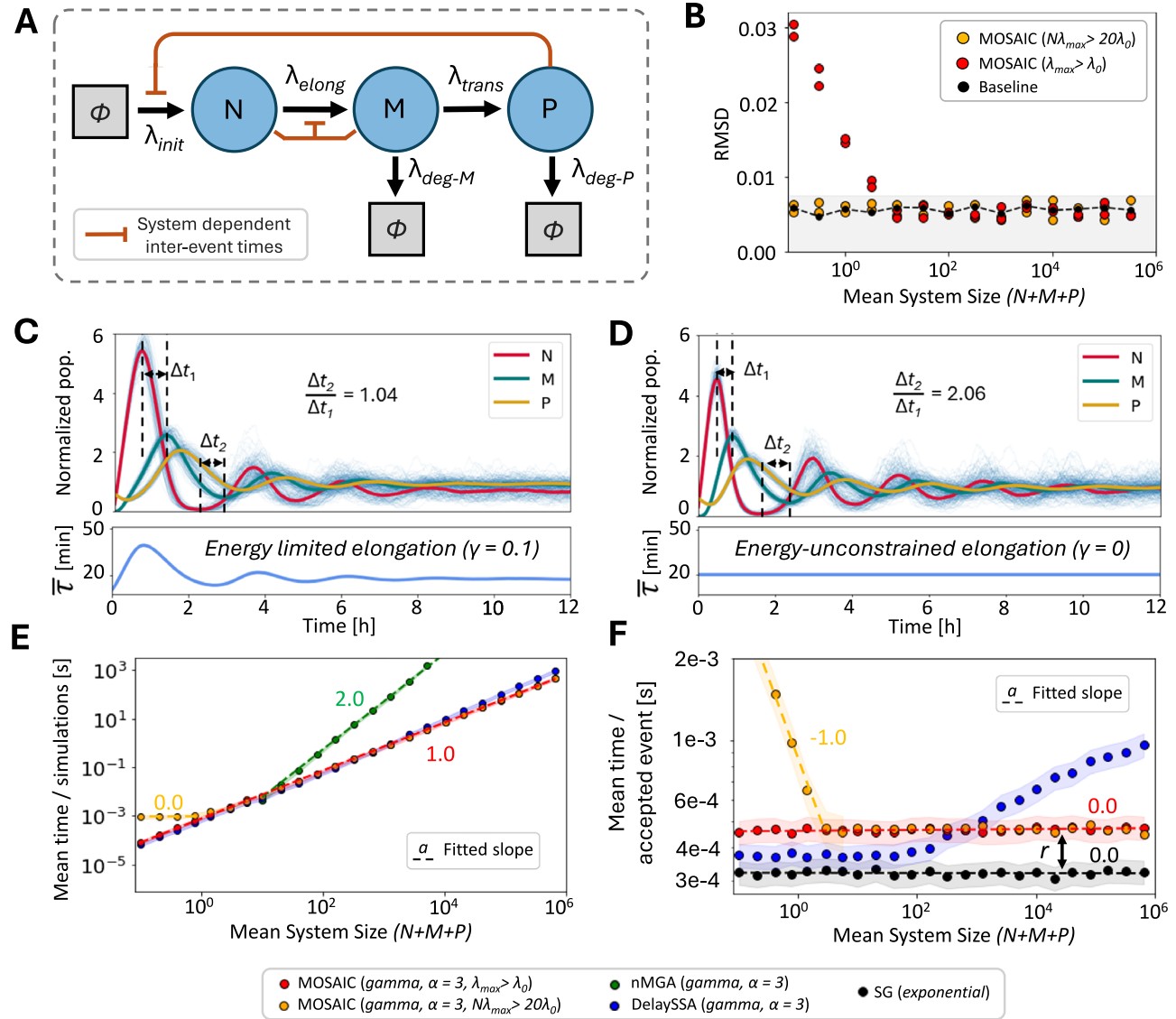

**Fig. 2 | MOSAIC accurately simulates Hes1 dynamics with state-dependent elongation delays. A** Schematic of the Hes1 negative feedback model. Nascent RNA (N) elongates until it matures into mRNA (M), which is translated into protein (P). Both mRNAs and proteins degrade at constant rates. Hes1 protein inhibits its own transcription, introducing a feedback loop that depends on the current system state. **B** Root mean square deviation (RMSD) between MOSAIC and DelaySSA trajectories as a function of mean system size ($\approx 25\beta$). Black dots show the baseline RMSD between two DelaySSA ensembles, while red dots compare MOSAIC against DelaySSA. Orange dots show MOSAIC with stricter $\lambda_{max}$ settings, further reducing error. Shaded regions indicate the 95% confidence interval expected between

identical simulations. **C, D** Population oscillations in nascent RNA, mature mRNA, and protein for energy-limited ($\gamma > 0$, (**C**)) and energy-unconstrained ($\gamma = 0$, (**D**)) elongation. Thin blue lines show single realizations, while thick lines show the average over 1000 simulations. Altered ratios of peak-to-trough delays ($\Delta t_2/\Delta t_1$) highlight the impact of state-dependent elongation times. **E** Average runtime per simulation as $\beta$ increases, showing near-linear scaling of MOSAIC compared to superlinear growth in alternative methods. **F** Runtime normalized by accepted events. MOSAIC maintains constant per-event cost, differing from standard Gillespie only by a rejection factor ($r \approx 1.6$ in this model).

Gamma-distributed waiting time, with the rate parameter further modulated by the number of nascent and mature RNAs ($N_N$ and $N_M$ respectively) to account for limited cellular resources:

$$\lambda_0 = \tau_0^{-1}\left(1 + \gamma \cdot \frac{N_N + N_M}{\beta}\right)^{-1}. \tag{8}$$

Here, $\tau_0$ defines the baseline mean delay, $\gamma$ controls resource sensitivity, and $\beta$ scales with system size. When $\gamma > 0$, elongation slows as transcripts accumulate, dynamically reshaping the waiting-time distribution. Importantly, such state-dependent delays cannot be captured by DelaySSA, which fixes delays at initiation, but are naturally accommodated by MOSAIC.

To benchmark accuracy, we first examine the case $\gamma = 0$ (no resource coupling). In this regime, elongation delays are fixed, and DelaySSA is exact, providing a natural ground truth for evaluating MOSAIC. MOSAIC trajectories closely track those of DelaySSA across a wide range of system sizes (Fig. 2B), with deviations vanishing as $\beta$ increases, consistent with theoretical error bounds (Eq. (6)). Stricter choices of $\lambda_{max}$ further reduce errors in small systems, at only modest additional computational cost (Supplementary Section B.4); we find here that increasing $\lambda_{max}$ above the true maximum rate is only beneficial for systems with fewer than about 20 processes. When $\gamma > 0$, elongation delays become context-dependent, producing marked changes in oscillatory behavior. In particular, oscillations in mRNA and protein populations lengthen

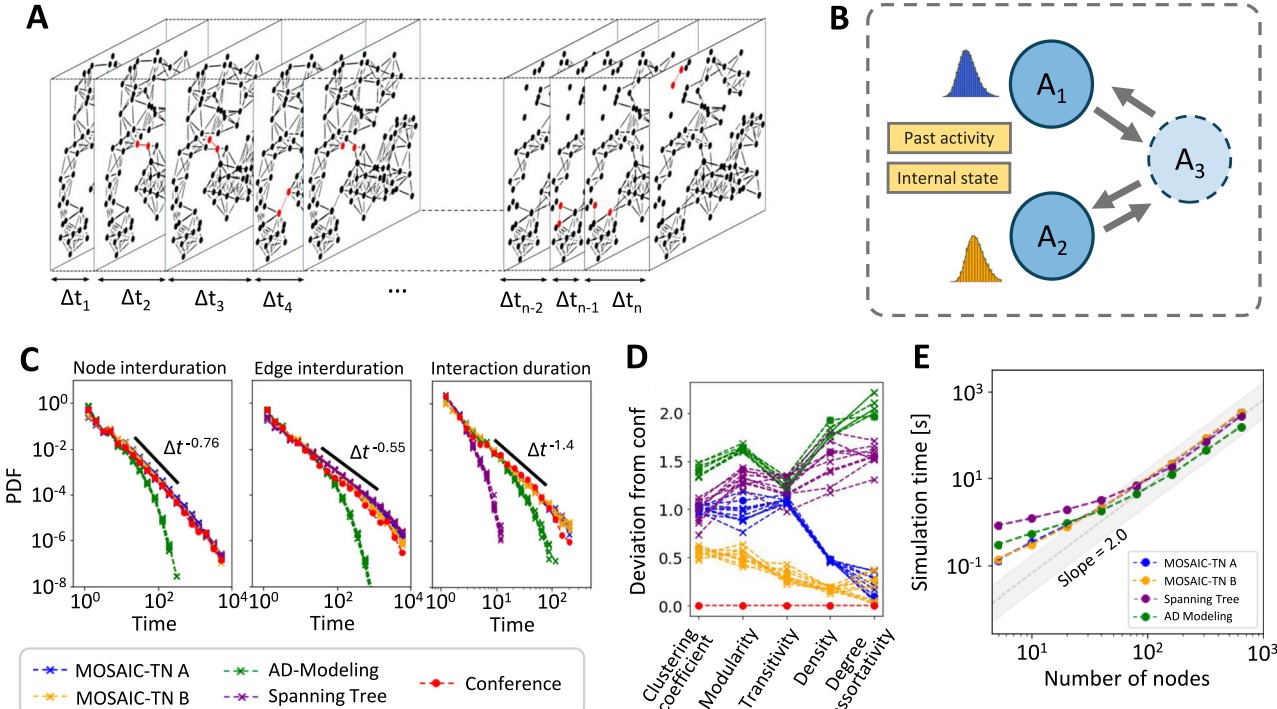

**Fig. 3 | MOSAIC-TN reproduces empirical social-temporal statistics with bursty node activity and interaction memory.** **A** Schematic of a temporal network simulated with MOSAIC-TN. At each time step, one link at most is updated; modified nodes and edges are shown in red. Rejected steps leave the network unchanged. The time increment $\Delta t$ is variable and computed according to Eq. (15). **B** Cartoon of the face-to-face interaction model. Each node tracks its own inter-event time distribution (IED), shown as blue and brown curves, and pairwise interaction probabilities depend on both intrinsic activity and past contacts. **C** Probability densities for temporal interaction statistics in conference data[58] and in simulated networks, averaged over 10 runs per model. Left: node interdurations, i.e. time intervals between successive interactions of the same node. Middle: edge

interdurations, i.e. time intervals between successive interactions of the same pair of nodes. Right: interaction durations, i.e. edge lifetimes. Power-law exponents fitted to the tails of the conference distributions are annotated in the panels. Two MOSAIC-TN variants are shown: MOSAIC-TN A (no partner preference) and MOSAIC-TN B (reinforced past interactions), alongside other baselines. **D** Absolute deviations of aggregated network metrics (clustering, modularity, transitivity, density, assortativity) from conference data. Deviations are normalized by the standard deviation across models; values are averaged over 10 simulations per model. **E** Computational runtime as a function of network size ($N_A$). For $N_A > 100$, all models exhibit $O(N_A^2)$ scaling (shaded region).

in period as elongation slows at high transcript abundance (Fig. 2C, D). This effect reflects competition for limited cellular resources, which in practice constrains the energy and machinery available for transcription. MOSAIC naturally captures these shifts by continuously updating reaction rates with the system state. In contrast, DelaySSA schedules delays at initiation and keeps them fixed, making it unsuitable for systems with dynamically evolving inter-event times (Table 1). These results demonstrate how MOSAIC extends Gillespie-like simulation to biologically realistic settings with state-dependent delays.

We next assess runtime scaling across $\beta$, which controls the average number of molecules (Fig. 2E, F). For comparison, we include the non-Markovian Gillespie algorithm (nMGA)[25], which extends the Gillespie framework to non-exponential waiting times by explicitly tracking all pending reactions. As expected, nMGA exhibits quadratic scaling, while DelaySSA incurs $O(\log N)$ cost per accepted event due to maintaining a priority queue of delays (Table 1). In contrast, MOSAIC achieves linear scaling with a constant per-event cost, differing from the standard Gillespie algorithm only by a multiplicative rejection factor $r$ (here $r \approx 1.6$). Thus, MOSAIC unites the flexibility to model dynamic, non-Markovian delays with the computational efficiency of classical stochastic simulation.

Together, these results demonstrate that MOSAIC enables realistic modeling of feedback-regulated processes, including transcription, signaling, and cell-cycle progression, where capturing non-exponential dynamics is critical for understanding system behavior.

## Application III: Non-Markovian temporal networks (MOSAIC-TN)

Temporal networks describe systems in which connections between nodes evolve over time, capturing both the timing and the duration of interactions (Fig. 3A). They provide a natural framework for studying dynamical processes such as opinion spreading[47,48], epidemic transmission[49,50], brain activity[51], and the diffusion of innovations[52]. Many such processes are inherently non-Markovian, characterized by bursty human activity, long inactive periods, and history-dependent dynamics that cannot be captured by memoryless exponential waiting times[53–55]. Temporal networks can be modeled according to two complementary perspectives: *edge-driven* or *node-driven*. In edge-driven models, each link $(i, j)$ has its own clock, and an interaction occurs when that clock fires[56,57]. In node-driven models, each node carries an activity process that generates partner interactions. When all clocks are exponential, these two perspectives are mathematically equivalent: the superposition of exponential edge clocks yields exponential node activity, and vice versa. In this Markovian setting, standard Gillespie methods apply seamlessly to either formulation.

For non-exponential IEDs, however, this equivalence breaks down (Supplementary Section C). Node activity statistics can no longer be obtained by simply combining edge clocks, and conversely, prescribing node-level IEDs does not translate into straightforward edge processes. Existing Gillespie extensions (DelaySSA, nMGA, Laplace Gillespie) remain fundamentally edge-centric, unable to explicitly capture heterogeneous node activity patterns or adapt to self-exciting and bursty dynamics. Yet many real-world temporal networks are

inherently node-driven. For instance, in social systems, individuals alternate between long inactive periods and short bursts of intense interaction[13,58]. In neuroscience, neurons generate spike trains with refractory periods and history-dependent firing[59]. In molecular biology, receptors and enzymes cycle through inactive and active states that shape their binding opportunities[60]. In all these examples, it is the *intrinsic dynamics of nodes*, rather than the edges, that primarily govern interaction statistics.

In this final use case, we show that MOSAIC can be extended to node-driven systems to capture these behaviors (MOSAIC-TN; Methods 3.4). Namely, we consider a temporal network where each node $A_i$ carries its own clock $t_i$, which records the time since the last interaction. Pairwise events are governed by a kernel rate $\Lambda_{ij}(t_i, t_j)$, which combines the activities of the two nodes and can incorporate history-dependent interaction terms. More generally, $\Lambda_{ij}(t_i, t_j)$ can be scaled by a global network-topology factor $H(i, j)$, allowing it to encode complex, non-independent relationships between nodes (e.g., higher-order structural effects beyond direct adjacency). At each iteration, two nodes are sampled according to the desired interaction topology (e.g., by drawing an edge from the underlying static network), and their interaction is accepted with probability $\Lambda_{ij}(t_i, t_j)/\Lambda_{\max}$, where $\Lambda_{\max}$ is an upper bound on all pairwise rates. As in MOSAIC, rejection sampling ensures $O(1)$ complexity per candidate event while allowing arbitrary node-level IEDs.

For a multiplicative kernel of the form $\Lambda_{ij} \propto \lambda_i(t_i) \cdot \lambda_j(t_j)$, interactions occur only when both nodes are concurrently available, since the product suppresses events whenever either node's intrinsic rate is low. This probabilistic synchronization couples node and edge dynamics, and in the limit where the number of links $N \to \infty$, the activity of each node converges to its intrinsic IED (Supplementary Section D.3). In this way, MOSAIC-TN generalizes Gillespie-style simulation to non-Markovian temporal networks, consistently linking individual node activity patterns to emergent interaction dynamics.

We next apply MOSAIC-TN to model face-to-face interactions recorded at the International Conference on Computational Social Science (IC2S2, 2017)[58]. Conferences provide a nearly well-mixed setting in which any two participants can, in principle, interact, so we model the system as a fully connected underlying contact graph and let temporal dynamics determine which edges are active at any given time. Empirical analyses show that both the *node interduration*, i.e. the time between successive interactions of the same individual, and the *interaction duration*, i.e. the time an edge persists, follow heavy-tailed distributions, well approximated by Pareto laws[20,58]. Accordingly, we fit Pareto distributions to the IC2S2 data and use the resulting parameters to define two kinetic channels (Methods 3.6): (i) $A_i + A_j \to [A_iA_j]$ with $\psi(t) \sim t^{-\alpha_A}$ for node interdurations, and (ii) $[A_iA_j] \to \varnothing$ with $\psi(t) \sim t^{-\alpha_\varnothing}$ for interaction durations (Fig. 3B,C).

To test the role of interaction memory, we implement two variants. In MOSAIC-TN A, all pairs interact with equal probability, independent of past history. In MOSAIC-TN B, partner selection is history-dependent: once $A_i$ is chosen, nodes $A_j$ with prior interactions are $w$ times more likely to be selected. This mechanism captures first-order temporal neighborhood effects, where past contacts increase the likelihood of repeated interactions. Remarkably, despite introducing only a single additional parameter, $w$, this history-based reinforcement substantially improves model performance. Optimizing $w$ to minimize deviations in network clustering, modularity, transitivity, density, and assortativity (Methods 3.6) yields markedly better agreement with empirical data on four of the five metrics compared to MOSAIC-TN A (Fig. 3D).

We then benchmark MOSAIC-TN against two established methods, the spanning-tree framework and activity-driven (AD) models. The spanning-tree approach[27] imposes specified node and edge IEDs but lacks flexibility in modeling interaction durations, leading to systematic underestimation of edge lifetimes (Fig. 3C). AD models[20,35]

provide greater flexibility but require extensive parameter tuning and inherently generate exponential tails, failing to reproduce the heavy-tailed node interdurations observed in empirical data (Fig. 3C). By contrast, MOSAIC-TN directly enforces arbitrary node-level IEDs while preserving exact event-driven timing, enabling it to simultaneously match both the interduration statistics and the distribution of interaction durations (Fig. 3C). In our instantaneous-rate framework, node and edge interdurations are mathematically related (Supplementary Section C), and MOSAIC-TN reproduces the empirically observed burstiness at both levels. Previous models that incorporated bursty node IEDs[28,61] treated interactions independently: nodes could display bursty activation while edge IEDs remained effectively memoryless. The spanning-tree framework[27] captures edge-level burstiness but at the cost of reduced flexibility. MOSAIC-TN unifies these advantages by enforcing bursty node IEDs, reproducing bursty edge interdurations, and retaining flexible, realistic interaction durations within a single coherent framework.

We next assess computational scaling with network size, where $N_A$ denotes the number of nodes in the social network (Fig. 3E). In this fully connected setting, the number of processes scales as $N \sim N_A^2$, since each pair of nodes can potentially interact. At smaller $N_A$, runtimes differ across methods: MOSAIC-TN A and B achieve the fastest execution, AD models followed by leveraging parallel updates, while the spanning-tree approach incur higher costs. As $N_A$ increased, all methods converge to the expected quadratic scaling $O(N_A^2)$, consistent with the combinatorial growth in interactions. Importantly, MOSAIC-TN preserves its constant per-event cost across the full range of network sizes, thereby matching the asymptotic efficiency of classical Gillespie-type approaches. Moreover, MOSAIC-TN A and MOSAIC-TN B remain indistinguishable, confirming that history-based reinforcement introduces no measurable computational overhead.

In summary, MOSAIC-TN combines the statistical precision of Gillespie-style simulation with the flexibility to impose arbitrary node-level IEDs and incorporate temporal memory. When applied to social contact data, it faithfully reproduces heavy-tailed interdurations, realistic edge lifetimes, and aggregated network statistics-outperforming both spanning-tree and AD models. Beyond social systems, this flexibility is essential for capturing complex real-world dynamics, including temporal neighborhood effects[62,63] and higher-order interactions[64-66], where nodes preferentially engage with neighbors of their neighbors[20,67,68]. Thus, MOSAIC-TN establishes a principled and scalable framework for simulating non-Markovian temporal networks, with broad applications ranging from human interaction patterns to neuronal firing and molecular binding.

## Discussion
Real-world systems often display memory effects and heterogeneity among agents and nodes, from transcriptional regulation to epidemic spreading[13,15,16]. These features are closely connected, as diversity in agents and their interactions naturally produces memory effects and non-exponential dynamics. MOSAIC introduces a unifying framework that bridges two frameworks long kept separate. Gillespie-style algorithms offer elegance, efficiency, and analytical rigor but assume statistical homogeneity among agents[17,25,26,69]. Agent-based approaches, in contrast, can encode arbitrary heterogeneity but typically sacrifice scalability and mathematical tractability[18,30,35]. MOSAIC combines the strengths of both: it retains the rigor and scalability of Gillespie-type methods while extending their flexibility to heterogeneous agents with distinct distributions, rates, and interaction preferences. This synthesis establishes MOSAIC as a general framework for systems in which memory effects and heterogeneity are both essential.

We have demonstrated MOSAIC's versatility across three distinct domains. First, in a system of B cell competition and evolution, MOSAIC efficiently models clonal selection driven by B cell receptor diversity. B cells with different antigen affinities compete for limited T

cell help, where only the highest-affinity B cells are rescued from apoptosis, allowed to divide, and may accumulate beneficial mutations. This competition is central to affinity maturation but also introduces algorithmic challenges, since each B cell must be represented with its own receptor-specific affinity. In standard Gillespie implementations, such heterogeneity leads to a combinatorial explosion of possible interactions. By contrast, MOSAIC uses a rejection-based scheme in which candidate interactions are sampled uniformly and only those meeting the strict affinity threshold are accepted. This avoids maintaining or updating a full list of competing pairs, enabling constant-time updates per event even in large heterogeneous populations, while faithfully reproducing experimentally observed patterns of clonal dominance during affinity maturation. Second, in an RNA transcription system with delayed negative feedback, MOSAIC captures state-dependent, non-exponential waiting times that classical delay-based algorithms cannot dynamically adjust. In this system, transcription initiation is followed by broadly distributed elongation times shaped by polymerase pausing and protein negative feedback that inhibits further initiation. By continuously updating reaction rates with the evolving system state, MOSAIC reproduces realistic oscillatory dynamics in both mRNA and protein abundances, consistent with experimental data, and does so without the restrictions of fixed-delay methods, which lock delays at initiation and prevent adaptation to changing cellular conditions. Third, in temporal social networks, MOSAIC-TN captures node-driven activity patterns in which individuals alternate between long inactive periods and short bursts of interactions. By directly enforcing arbitrary node-level inter-event distributions and incorporating memory in partner selection, it reproduces heavy-tailed inter-event times, realistic edge lifetimes, and higher-order network features. In doing so, it outperforms spanning-tree and activity-driven models, which either underestimate interaction durations or rely on exponential assumptions. Together, these use cases show that MOSAIC unifies heterogeneity and memory in a single scalable framework, recovering empirical features that existing methods approximate poorly or fail to capture altogether.

MOSAIC is most efficient for large systems with moderate heterogeneity and broad inter-event distributions. Its advantage diminishes when rejection rates are high or populations extremely small (Supplementary Section B.5), where delay-based approaches may remain preferable[24]. A further limitation arises when inter-event time distributions become highly peaked or nearly discrete (e.g., concentrated on multiples of a fixed time step): in this regime, representing the dynamics via a continuous-time hazard requires very large upper bounds in the thinning procedure, which leads to heavy rejection and poor efficiency. For strictly time-discrete models with atomic event times, dedicated discrete-time or DelaySSA-style algorithms remain better suited, and MOSAIC should be viewed only as a continuous-time approximation rather than the method of choice.

More generally, all stochastic simulation frameworks face challenges when applied to systems with very large numbers of reactants and reaction channels. In such cases, analytical simplifications or surrogate approaches, such as training neural networks on limited simulations to generate large-scale ensembles[70], can reduce computational cost. Another fundamental difficulty arises when reaction propensities span several orders of magnitude: the dominance of high-frequency events can make rare events prohibitively expensive to observe, as noted in RNA transcription studies[71]. Piecewise-deterministic Markov process (PDMP) models provide an effective alternative by simulating rare events stochastically while handling frequent events deterministically, leading to major speedups in diverse applications[71–74]. Hybrid extensions of MOSAIC that combine these ideas represent a natural next step.

In our temporal-network examples, we focused on a fully connected topology and networks of up to $10^3$ agents, primarily to keep the exposition and comparison with SSA-based baselines as simple and

transparent as possible. This choice is made for illustration rather than realism, and many applications will naturally involve constrained, sparse, but large contact networks[75,76]. Within MOSAIC-TN, such constrained topologies can be incorporated straightforwardly: candidate interactions can be restricted to edges of an arbitrary static network, and both the pair-selection mechanism and the pairwise kernel $\Lambda_{ij}(t_i, t_j)$ can encode dynamic topological structure. In that setting, the number of processes would scale with the number of admissible edges rather than the square of the population size, opening the door to larger systems on sparse graphs. Nonetheless, our current implementation is not an HPC framework and is not engineered for national-scale, distributed-memory simulations. Large-scale tools such as EpiFast[32] and EpiHiper[34] target a different regime, namely discrete-time dynamics on massive sparse networks. Systematically exploring sparse and application-specific topologies, and developing optimized implementations for that regime, is therefore an important direction for future work[77].

In modeling interactions, we focused here on the case where both nodes must be simultaneously available, which is natural for undirected contact networks such as face-to-face encounters or synaptic transmission. This multiplicative formulation coordinates node and edge activity and is well suited for systems where neither participant can sustain an interaction independently. Importantly, MOSAIC is not restricted to this choice. In settings where interactions are driven primarily by one participant, such as epidemic spreading, email exchanges, or message transmissions, an additive formulation of the pairwise rate, $\Lambda_{ij} \propto \lambda_i(t_i) + \lambda_j(t_j)$, can be used instead (Supplementary Section D.3). The framework therefore accommodates both synchronous and sender-driven dynamics, underscoring its flexibility across different classes of networks.

Looking ahead, MOSAIC provides a foundation for extending stochastic simulation to increasingly complex and large real-world systems. For instance, hybrid deterministic-stochastic schemes can further improve efficiency in systems with widely separated time-scales. Coupling MOSAIC with differential-equation models, i.e. ordinary differential equations for population averages or partial differential equations for spatially structured processes, can enable true multiscale integration, linking individual variability to emergent population-level outcomes. On the application side, MOSAIC provides a versatile framework for modeling patient-specific variability in personalized medicine, capturing behavioral heterogeneity in epidemic control, and simulating adaptive dynamics in synthetic biology, neuroscience, and financial markets.

In conclusion, MOSAIC demonstrates that mechanistic realism and computational efficiency need not be traded off. By unifying memory and heterogeneity in a scalable stochastic framework, it establishes a versatile foundation for modeling complex systems and opens powerful new avenues for understanding how diversity and memory drive collective dynamics.

## Methods
### MOSAIC implementation
The MOSAIC implementation was optimized around two key aspects to reduce computational cost. First, instead of updating the elapsed time $t_j$ since the last reaction for every process at each iteration, we only store the last reaction timestamp $\tau_j$ for process $j$ and compute $t_j = T - \tau_j$ when required. Second, the calculation of $\lambda_{max}$ (the upper bound on instantaneous reaction rates) is avoided at every step by exploiting the structure of the inter-event time distributions. Four scenarios can be distinguished:

- *Bounded rates.* For distributions with a bounded instantaneous rate (e.g., Pareto, log-normal, Cauchy, delayed exponential, Gamma with $\alpha > 1$), $\lambda_{max}$ can be set as the fixed upper bound.
- *Unbounded, increasing rates.* For distributions with an unbounded rate that increases monotonically with time (e.g., normal, Weibull

with $\alpha > 1$), $\lambda_{\max}$ is determined by the reactant that has not reacted for the longest time, $\lambda_{\max} = \lambda(t_{\max})$. The bound only needs updating when this reactant undergoes a reaction.

- *Unbounded, decreasing rates.* For distributions with an unbounded rate that decreases monotonically with time (e.g., Weibull with $\alpha < 1$, Gamma with $\alpha < 1$, or power-law forms), $\lambda_{\max}$ is determined by the reactant with the shortest waiting time. Since this can yield excessively large $\lambda_{\max}$, MOSAIC is not recommended for such cases, where the Laplace Gillespie algorithm[26] is more efficient.

- *Arbitrary rates.* For distributions with arbitrary shapes (e.g., numerical fits from experimental data), the user should precompute bounds on $\lambda(t)$ within intervals. This reduces the problem to one of the three categories above for each interval.

## B-cell affinity maturation model

During an immune response, B cells and T cells migrate within germinal centers (GCs)[78], where they encounter one another through random motion and transient contacts. B cells that successfully bind antigen present fragments to T helper cells, which in turn provide survival and proliferation signals. Because the number of available T cells is limited, B cells compete for this help, and clones with higher-affinity receptors are more likely to be rescued from apoptosis, to proliferate, and to accumulate affinity-enhancing mutations. This competitive selection process is the basis of affinity maturation, and we capture it here with a minimal stochastic model consisting of five reaction processes:

- *B cell binding to T cells:* $B + T \xrightarrow{\lambda_{BT}} [BT]$. A B cell encounters and initiates an interaction with a T cell. Here, we assume that T cells are all identical.

- *B-cell competition for T-cell help:* $[B_1 T] + B_2 \xrightarrow{\lambda_{BT}} (IP)[B_2 T] + B_1$. Even if a cell $B_1$ is already interacting with a T cell and receiving survival signals ($B_1 T$), it can be displaced by a new cell $B_2$ with a higher-affinity receptor, i.e. affinity($B_2$) > affinity($B_1$). This competition is central to driving affinity maturation in B cells. (IP) in the reaction equation denotes that the process incorporates the individual properties (IP) of each B cell, accounting for variations in receptor affinity, and resulting in different propensities, or rates, for each existing (B, T) pair.

- *B-cell apoptosis:* $B \xrightarrow{\lambda_{apop}} \varnothing$. B cells undergo apoptosis if not rescued by T cells. This mechanism eliminates non-competitive B cells from the system.

- *B-cell spontaneous unbinding from T cells:* $[BT] \xrightarrow{\lambda_{unbind}} B_{div} + T$. After receiving sufficient survival signals, B cells detach from their T cell and prepare for further divisions.

- *B-cell division:* $B_{div} \xrightarrow{\lambda_{div}} B + B$. Selected B cells undergo division, producing two daughter cells. Let $B_p$ and $B_d$ represent a parent and a daughter B cell, respectively. After each division event, the affinity change of the daughter cell $B_d$ is computed as follows:

$$\Delta_{aff} = \text{affinity}(B_d) - \text{affinity}(B_p) = \frac{u - \text{affinity}(B_p)}{\beta}, \quad (9)$$

where $u$ is a uniformly distributed random variable in the range [0, 1], and $\beta$ is a scale parameter that determines the magnitude of the affinity change. This formulation makes it more challenging for cells with higher affinity values to further improve their affinity (Supplementary Fig. S5A) In our simulations, B-cell affinities start at 0 and evolve dynamically through successive divisions. After 50 days, affinity values reach approximately 0.7 ± 0.05 (Supplementary Fig. S5B), reflecting the gradual accumulation of affinity-enhancing mutations over time.

We initialized the system with $N_B = 1000$ B cells and $N_T = 10$ T cells. Simulations were run for 50 days using the following parameters (from[11]): $\beta = 10$, $\lambda_{BT} = 0.146\ h^{-1}$, $\lambda_{unbind} = 2\ h^{-1}$, $\lambda_{apop} = 0.084\ h^{-1}$, and $\lambda_{div} = 0.134\ h^{-1}$. For the reaction involving B-T competition, we set $\lambda_{\max} = \lambda_{BT}$ and kept this choice fixed throughout the simulation.

## Hes1 transcription model with delayed negative feedback

We modeled RNA transcription and protein synthesis in the *hes1* system, which is known to exhibit oscillatory behavior driven by delayed auto-inhibition[42–44]. The protein Hes1 represses the transcription of *hes1* RNA, generating a negative feedback loop that has been widely studied as a prototype for regulatory circuits with memory[79,80]. The system was represented by three species: nascent RNA ($N$), mature mRNA ($M$), and protein ($P$), with populations denoted $N_N$, $N_M$, and $N_P$. The dynamics of *hes1* transcription and translation were described by five stochastic reactions, each assigned an IED that specifies the timing of events. Building upon the work of Monk[43], where parameters were fit from experimental data, we specified the reactions as follows:

- *RNA transcription initiation:* $\varnothing \xrightarrow{\beta \cdot G(N_P)} N$. The initiation of *hes1* transcription is modeled with an instantaneous rate that decreases with the amount of Hes1 protein $N_P$[44]. Here, $G(N_P)$ is a Hill function with coefficient $h = 4.1$:

$$G(N_P) = \frac{1}{1 + \left(\frac{N_P}{\beta}\right)^h}, \quad (10)$$

where $\beta$ is the protein concentration required for half-maximal repression.

- *nRNA elongation:* $N \xrightarrow{\tau \sim \mathcal{G}(\lambda_0, \alpha_{elong})} M$. Elongation of an RNA molecule typically takes 15–20 min. We modeled this delay using a Gamma distribution $\mathcal{G}$ with mean $\tau_0 = 20$ min, shape parameter $\alpha_{elong} = 3$, and rate $\lambda_0 = 1/\tau_0$, as parametrized in Supplementary Section A.1. The Gamma distribution captures the variability of elongation times, which are often heavy-tailed due to pausing events[45,46]. To incorporate resource limitation, we used a generalized form[25]:

$$\lambda_0 = \tau_0^{-1}\left(1 + \gamma \cdot \frac{N_N + N_M}{\beta}\right)^{-1}, \quad (11)$$

where $\gamma$ controls the effect of limited resources, and $N_N$ and $N_M$ are normalized by $\beta$. Setting $\gamma = 0$ recovers a constant elongation rate.

- *Protein synthesis:* $M \xrightarrow{\lambda_{trans}} M + P$. Mature mRNA molecules are translated into protein with rate $\lambda_{trans} = 0.01\ min^{-1}$. Translation is treated as effectively instantaneous, since completion typically occurs within one minute[44]. Each mRNA can be reused multiple times before degradation.

- *RNA degradation:* $M \xrightarrow{\lambda_{deg-M}} \varnothing$. mRNA molecules degrade with rate $\lambda_{deg-M} = 0.029\ min^{-1}$. This process was modeled as a Markovian exponential decay, consistent with experimental observations[81].

- *Protein degradation:* $P \xrightarrow{\lambda_{deg-P}} \varnothing$. Proteins degrade with rate $\lambda_{deg-P} = 0.031\ min^{-1}$, modeled as a Markovian exponential decay process.

For the nRNA elongation reaction, the waiting time $\tau$ follows a Gamma distribution $\mathcal{G}(\lambda_0, \alpha_{elong})$ which we model in MOSAIC with the instantaneous rate

$$\lambda_{elong}(t) = \frac{\psi_{elong}(t)}{1 - F_{elong}(t)} = \frac{(\alpha_{elong}\lambda_0)^{\alpha_{elong}} t^{\alpha_{elong}-1} e^{-\alpha_{elong}\lambda_0 t}}{\Gamma\left(\alpha_{elong}, \alpha_{elong}\lambda_0 t\right)} \leq \lambda_{\max}^{(elong)} = \alpha_{elong}\lambda_0, \quad (12)$$

where $\Gamma(\cdot,\cdot)$ denotes the upper incomplete gamma function, and $\lambda_{\max}^{(\text{elong})}$ the global upper bound for the elongation channel in MOSAIC. For small systems, where $N = N_N + N_M + N_P$ is modest, we also evaluated a more conservative choice to ensure a smaller effective time step $\Delta t$ (Supplementary Section B.4), by enforcing

$$N\lambda_{\max}^{(\text{elong})} \geq 20\lambda_0, \quad \text{which implies an average time step } \mathbb{E}[\Delta t] \leq \frac{1}{20\lambda_0}. \tag{13}$$

We assessed MOSAIC's accuracy under the specific condition of $\gamma = 0$, where the distribution of RNA elongation inter-event times remains unaffected by the RNA molecule population (energy-unconstrained elongation). We executed MOSAIC and DelaySSA across $N_{\text{sim}} = 10^4/\sqrt{\beta}$ simulations and compared the root mean square deviation (RMSD) of normalized averaged populations for different values of the scale parameter $\beta$. As a baseline, we used the mean RMSD between two independent realizations of DelaySSA simulations. The number of simulations was scaled by $1/\sqrt{\beta}$ to compensate for the increased stochasticity in smaller systems.

## The MOSAIC-TN algorithm

MOSAIC-TN extends the rejection-based Gillespie framework to node-driven temporal networks, where each node follows its own IED and carries an internal clock $t_i$ encoding its elapsed time since the last event. Interactions are constrained by a prescribed topology. Let $N_A$ denote the number of nodes, and let $\mathcal{E}$ denote the set of admissible node pairs $(i,j)$ consistent with the desired interaction graph. We write $N = |\mathcal{E}|$ for the number of such pairs; in the fully connected case, $N = N_A(N_A - 1)/2$. The pairwise rates $\Lambda_{ij}(t_i, t_j)$ depend on the internal times of nodes $i$ and $j$. The procedure is as follows:

(i) Set $\Lambda_{\max}$ as an upper bound on all possible node–node interaction rates over admissible pairs:

$$\Lambda_{\max} \geq \max_{(i,j)\in\mathcal{E}} \Lambda_{ij}(t_i, t_j). \tag{14}$$

(ii) Compute the time increment to the next candidate event using $\Lambda_{\max}$. Draw a uniform random variable $u \in \mathcal{U}[0,1]$ and advance time by

$$\Delta t = \frac{\ln(1/u)}{N\Lambda_{\max}}, \tag{15}$$

(iii) Select a candidate pair $(i,j)$ for the next event by sampling uniformly from the admissible set $\mathcal{E}$. Thus, each pair has probability

$$p_{ij} = \frac{1}{N}, \ (i,j) \in \mathcal{E}. \tag{16}$$

(More general proposal distributions over $\mathcal{E}$ can also be used, provided $\Lambda_{\max}$ remains a valid global bound, as discussed in Supplementary Section G)

(iv) Accept the event with probability

$$p_{\text{accept}} = \frac{\Lambda_{ij}(t_i, t_j)}{\Lambda_{\max}}, \tag{17}$$

and update the states and clocks of nodes $i$ and $j$ accordingly. If the event is rejected, the step is treated as a null event: time advances by $\Delta t$, but the node states remain unchanged.

## Practical implementation for Pareto-distributed activity

For social interactions we model node activity with a Pareto law $\mathcal{P}(\lambda_A, \alpha_A)$, consistent with empirical human contact data[20,58]. The IED of

node activity is parametrized as

$$\psi(t) = \frac{\alpha_A t_{\min}^{\alpha_A}}{t^{\alpha_A + 1}}, \ t \geq t_{\min}, \ \text{with instantaneous rate } \lambda(t) = \frac{\alpha_A}{t}. \tag{18}$$

We define the characteristic interaction rate $\lambda_A$ as the reciprocal of the median inter-event time,

$$t_{\text{med}} = t_{\min} 2^{1/\alpha_A}, \ \lambda_A = \frac{1}{t_{\text{med}}} = \frac{2^{-1/\alpha_A}}{t_{\min}}. \tag{19}$$

This definition remains valid for all $\alpha_A > 0$, including heavy-tailed regimes where the mean diverges.

For MOSAIC-TN, we consider a fully connected underlying network and adopt a multiplicative pairwise rate,

$$\Lambda_{ij}(t_i, t_j) = \frac{\lambda_i(t_i)\lambda_j(t_j)}{(N_A - 1)\lambda_A}, \tag{20}$$

which guarantees that as $N_A \to \infty$, each node's IED converges to its prescribed Pareto form (Supplementary Section D.3). Finally, because the Pareto hazard rate $\lambda(t)$ decreases monotonically after $t_{\min}$, we can set a constant global upper bound for the entire simulation as

$$\lambda_{\max} = \frac{\alpha_A}{t_{\min}}, \ \Lambda_{\max} = \frac{\lambda_{\max}^2}{(N_A - 1)\lambda_A}. \tag{21}$$

To capture temporal neighborhood effects, we also implement MOSAIC-TN B, where partner nodes are drawn non-uniformly: once $A_i$ is selected, previous partners $A_j$ are $w$ times more likely to be chosen. This modification reduces rejection overhead at large $w$ and remains valid for Pareto and Weibull IEDs (Supplementary Section G). By directly sampling more probable interactions, MOSAIC-TN B avoids the sharp increase in rejected events that would otherwise occur (e.g., nearly tenfold for $w = 9.4$ compared to the standard MOSAIC-TN).

## Modeling and evaluation of temporal social networks

Face-to-face interactions were modeled using two kinetic channels parameterized by Pareto-distributed inter-event times. First, encounters between two agents $A_i$ and $A_j$ were represented as

$$A_i + A_j \xrightarrow{\mathcal{P}(\lambda_A, \alpha_A)} A_i + A_j + [A_iA_j], \tag{22}$$

where $[A_iA_j]$ denotes the formation of an interaction edge. Importantly, the agents themselves are not consumed by this process and remain available to initiate further interactions, so multiple simultaneous edges per node are possible. This process defines the *node interduration*, corresponding to the time between successive interactions of a given agent. Parameters were estimated by fitting a Pareto distribution to the observed node interduration distribution in the IC2S2 conference data[58], yielding $\alpha_A = 0.76$ and $\lambda_A = 0.4$. Second, interaction termination was modeled as

$$[A_iA_j] \xrightarrow{\mathcal{P}(\lambda_\varnothing, \alpha_\varnothing)} \varnothing, \tag{23}$$

corresponding to the removal of an edge. A Pareto fit to the empirical distribution of interaction durations gave $\alpha_\varnothing = 1.4$ and $\lambda_\varnothing = 0.61$.

Simulations were run with $N = 274$ nodes over $T = 7249$ time units, consistent with the IC2S2 dataset. To evaluate model fidelity, we compared simulated networks against the empirical data using five aggregated network metrics: clustering coefficient[82], modularity[83], transitivity[84], network density[85], and degree assortativity[86]. After each simulation, pairwise interaction durations were summed, log-transformed, and normalized by their maximum to obtain an *interaction*

*score*. A graph was then constructed by connecting pairs of individuals with an interaction score greater than 0.4. Community detection was performed using the Louvain algorithm[87] with a resolution parameter of 1. Network metrics were computed in `networkx` (v2.8.8) using `nx.average_clustering`, `nx.algorithms.community.quality.modularity`, `nx.transitivity`, `nx.density`, and `nx.degree_pearson_correlation_coefficient`.

### AD modeling and spanning tree temporal network

For AD modeling temporal networks, we utilized the V9 network from Le Bail et al.[20], identified as the highest-performing model. Optimized simulation parameters for the conference dataset, along with the code for implementing AD modeling, were sourced from https://github.com/DidierLeBail/Temporal-networks-PhD-code. The conference dataset is labeled as `conf17` within this repository.

Spanning Tree temporal networks were generated by assigning node activity rates based on a Pareto distribution fitted to the conference data. The repository https://github.com/anzhisheng/Temporal-networks-by-spanning-trees was used for this purpose. Before generating the spanning trees, a Barabási-Albert graph[88] was constructed with $m = N/2$ neighbors per node, reflecting the observed conference data trend where nodes interact on average with half of the population at least once.

## Data availability

This study used previously published datasets on germinal center B-cell maturation[39] and conference interactions[58]. These data are available from the original publications and their associated repositories. All simulation data required to reproduce the findings are generated from the model definitions and parameters reported in the Methods.

## Code availability

The MOSAIC and DelaySSA Python implementations, together with the code and data needed to reproduce all figures, have been deposited on Zenodo at https://doi.org/10.5281/zenodo.18346965.

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

## Acknowledgements

The authors thank Jonathan Karr, Farshid Jafarpour, Srividya Iyer-Biswas, and Peter Ashcroft for their valuable suggestions. This research was supported by the COSMIC European Training Network, funded by the European Union's Horizon 2020 research and innovation program under grant agreement No 765158 and from the Swiss National Science Foundation (Sinergia grant CRSII5 193832).

## Author contributions

A.P. designed and implemented the MOSAIC and MOSAIC-TN frameworks, performed all simulations and benchmarks, and wrote the manuscript. N.B. and M.R.M. supervised the project and contributed to writing and editing the manuscript; M.R.M. additionally contributed to the design of MOSAIC. M.P. contributed to the implementation of MOSAIC and assisted with simulations. D.B. advised on dataset selection and the activity-driven modeling framework for social interactions and clarified terminology.

## Competing interests

The authors declare no competing interest
