## [Transparent Peer Review file · Nature Communications]

Unifying non-Markovian Dynamics and Agent Heterogeneity in Scalable Stochastic Networks

Corresponding Author: Dr Aurélien Pelissier

Version 0:

Reviewer comments:

Reviewer #1

(Remarks to the Author)

In this paper, the authors generalize the standard Markovian Gillespie algorithm to a non-Markovian version that accommodates a broad range of (memory-dependent) inter-event time distributions and accounts for individual heterogeneity. They then show the applicability of this method through three examples: reproducing B-cell affinity maturation, modeling dynamical processes with memory effects, and capturing bursty activity patterns observed in empirical temporal networks. This approach can handle more complicated modelling scenarios, while keeping the computational cost reasonable.

Overall, I think the paper is well written. I have a few questions:

1. The author mentioned that when Δt approaches 0, the algorithm recovers the classical renewal process with the given inter-event time distribution. This statement seems to assume the inter-event time distribution is continuous. How does the approach behave for time-discrete stochastic processes? that is, when the activation only occurs at positive integral moments. Can the proposed algorithm be directly applied in such cases?
2. As I understand it, one of the importance of this paper is to provide an efficient way to simulate a non-Markovian multi-agent system with lower computational cost than directly simulating the renewal process for each agent, while keeping a high accuracy with appropriate parameter choices. But when the distribution is discrete, the simulation based on renewal processes only need to determine agents' states at integral time points. Then an associated question, in this case, is that would the proposed approach require more sampling than the traditional approach and thus higher computational cost?
3. I don't think the author correctly implement the spanning-tree method for constructing temporal networks. The key advance of this method is its ability to simultaneously reproduce burstiness in node and edge activities, and the corresponding exponents for nodes and the edge are adjustable according to the assigned activity pattern, which is exactly opposed to the statement in line 242.
4. There are several limitations in using the proposed method to generate temporal networks. For instance, first, the authors assume each pair of agents has a chance to interact at some point, but many real-world temporal networks have an underlying topology that constrains the physical interactions between individuals, such as online social networks or transportation networks. Some individuals can never interact because they are not connected in the underlying topology. Second, the difficulty of generating temporal networks matched with empirical observations is the strong correlations between node and edge activations imposed by these underlying topologies. Mathematically, such a system should be modeled as a joint renewal process involving the activities of all nodes and edges, which cannot be decoupled into independent processes for each pair of nodes. But the proposed approach, at each time, simply samples two random individuals and determines their activations, ignoring the fact that the activity of these two nodes and the edge is affected by the other nodes and edges. In summary, I am fine with using the construction of temporal networks as an example, but there are clearly several limitations that the authors should acknowledge, for instance, this method only works for well-mixed underlying topologies.

(Remarks on code availability)
I confirm the code is available.

Reviewer #2

(Remarks to the Author)

(Remarks on code availability)
I confirm the code is available.

Reviewer #3

(Remarks to the Author)

The authors developed a stochastic simulation approach called MOSAIC (the abbreviation for modeling of stochastic agents with individual complexity), which can unify non-Markovian dynamics and agent heterogeneity. This approach extended the conventional Gillespie framework to non-Markovian systems, and also generalized the standard Gillespie algorithm by embedding heterogeneity directly at the level of individual agents. I think that a key of the approach is that the authors introduced the rejection probabilities of the involved events to reduce the computational complexity of non-Markovian systems. In addition, by only tracking a global maximum rate, the MOSAIC can efficiently scale to large populations and the interactions between multiple agents, while capturing diverse inter-event waiting-time distributions and individual-specific properties. Therefore, I think that the MOSAIC can be a general tool for stochastic simulation of various possible Markovian or non-Markovian systems across multidisciplinary. I appreciate the idea of this study and check all the mathematical equations, finding that they are correct. This manuscript is well written and organized. In a word, I would like to recommend it for acceptance. However, I have the following concerns and comments:

Three concerns:

1. I think that in the MOSAIC, “N” is a key quantity to be determined since it directly influences the convergence of the approach. What the “N” represents seems unclear: It seems to me that it represents the number of “reaction channel” in some places, e.g., on line 312, but “the product of the largest possible molecular numbers of reaction species in other places, e.g., $N = N_A * N_B$ ” on lines 55-56. If it is the former, then “N goes to infinitely” does not hold in some cases, e.g., gene-expression systems in general involves few reactions, and in this case, Δt would not tend to zero unless λ_{max} is chosen to be very large. I suggest that the authors clearly describe how the “N” is determined and how the λ_{max} is chosen. In addition, the rate λ_j (on line 71) does not depend on the system’s state?
2. The algorithmic procedure described on lines 70-77 is difficult to understand (but I know that this procedure only described the algorithm that the time advances only by Δt) (one single time step). What is the relationship between the elapsed time t_j since the last event of the j th process and the time increment Δt ? The authors did not clearly show how the time advances to arrive at the moment t_j by increasing the time increment Δt . Is the elapsed time t_j since the last event of the j th process known? I think that clarifying this point is very important for understanding the MOSAIC.
3. In the case that the “N” tends to infinity, the number of the loops in the algorithmic procedure would be very large. Thus, the MOSAIC would not be applied in the case of very large “N”. Is my understanding correct? In addition, it seems to me that the MOSAIC does not consider the reaction system’s state, unlike the standard Gillespie algorithm which explicitly considers the system’s state. I cannot understand this point. Can you clarify it or give some interpretations?

Minor comments:

1. Applications I and II, although representative, are actually small systems. For each of these systems, how is the λ_{max} chosen and what size?
2. On lines 136-137, the description “In the standard Gillespie algorithm, this requires tracking $N = N_B * N_T$ reaction channels” seems incorrect. Regarding “reaction channel”, the common understanding is that one reaction corresponds to one reaction channel. Application III is a “good” example since it can better interpret the implement of MOSAIC, and I think that this example is interesting.
3. For a general biochemical reaction network, the authors [JJ Zhang, TS Zhou. Markovian approaches to modeling intracellular reaction processes with molecular memory. PNAS 116, 47:23542-23550(2019)] used general waiting-time distributions to model the occurrence of inter-events. I suggest that the authors of this manuscript cite that paper.
4. The abstract seems a little long, and I suggest that the authors shorten it so that it is not more than 200 words.

(Remarks on code availability)
no remarks

Reviewer #4

(Remarks to the Author)

(Remarks on code availability)
Attached as PDF

Version 1:

Reviewer comments:

Reviewer #1

(Remarks to the Author)

The authors have answered my comments to my satisfaction. I don't have any further questions. I think the article is now ready for publication.

(Remarks on code availability)

Reviewer #2

(Remarks to the Author)

(Remarks on code availability)

Reviewer #3

(Remarks to the Author)

The authors have made revisions according to my last review report, and in particular, they have fixed my three concerns. After having read all the materials related to their re-submission, I am satisfied with the current version and in particular with their revisions. I have no more comments. In a word, I would like to recommend this manuscript for acceptance.

(Remarks on code availability)

Reviewer #4

(Remarks to the Author)

I have read with interest the revised manuscript and the response document written by the authors. The authors have clearly stated the limitations of the approach for certain regimes and for discrete time simulations. I have no further comments and believe the paper is suitable for publication in Nature Communications.

(Remarks on code availability)

Summary

We thank the reviewers for their careful reading of our manuscript and for the many constructive suggestions. The feedback helped us correct several inaccuracies and substantially clarify both the scope and the presentation of MOSAIC and MOSAIC-TN.

Conceptually, we have sharpened the presentation of our modeling framework and its assumptions. We now (i) clearly define the number of processes N as determined by the model structure rather than as a tunable parameter, (ii) explain the role of the global rate bound λ_{\max} as the key accuracy–efficiency parameter, and (iii) describe in detail how elapsed times and state dependence are tracked in MOSAIC. We also clarify the limitations of our approach for highly peaked or genuinely discrete-time inter-event distributions and contrast MOSAIC with discrete-time or DelaySSA-style methods in that regime.

For temporal networks, the reviews prompted two substantive improvements. First, we now include edge interdurations as an explicit observable, alongside node interdurations and interaction durations, and we have revised Figure 3C accordingly. This makes the relationship between node and edge-level temporal statistics more transparent and confirms that both spanning-tree and MOSAIC-TN reproduce the expected bursty interdurations. Second, we have revised the presentation of MOSAIC-TN to emphasize that the framework is compatible with arbitrary underlying topologies and non-uniform pairing rules, and we provide a new Supplementary Section describing how to constrain interactions to a given network.

Finally, we have refined our discussion of efficiency and scope, clarifying where MOSAIC offers practical advantages, explicitly acknowledging its limitations for extremely large systems, and situating it more carefully relative to existing SSA, delay, and HPC/agent-based modeling (ABM) frameworks. We have also incorporated the suggested references and shortened and streamlined the abstract and main text. Below, we provide a detailed, point-by-point response to each remark.

Reviewer 1:

In this paper, the authors generalize the standard Markovian Gillespie algorithm to a non-Markovian version that accommodates a broad range of (memory-dependent) inter-event time distributions and accounts for individual heterogeneity. They then show the applicability of this method through three examples: reproducing B-cell affinity maturation, modeling dynamical processes with memory effects, and capturing bursty activity patterns

observed in empirical temporal networks. This approach can handle more complicated modelling scenarios, while keeping the computational cost reasonable. Overall, I think the paper is well written. I have a few questions:

1. The author mentioned that when Δt approaches 0, the algorithm recovers the classical renewal process with the given inter-event time distribution. This statement seems to assume the inter-event time distribution is continuous. How does the approach behave for time-discrete stochastic processes? that is, when the activation only occurs at positive integral moments. Can the proposed algorithm be directly applied in such cases?

A purely discrete inter-event distribution (i.e., events only at integer times) cannot be represented exactly within MOSAIC/REGIR or any Gillespie-style hazard-based SSA. Mathematically, such distributions correspond to Dirac masses, which would require **zero rate everywhere and an infinite instantaneous rate at the discrete activation times**. This would in turn require an **infinite upper bound** (λ_{\max}) in the rejection procedure. In practice one can only approximate this by using a very large finite λ_{\max} , which typically leads to heavy rejection. For this reason, exact simulation of a truly discrete-time process is better handled by a dedicated discrete-time or DelaySSA-style queue, whereas MOSAIC is designed for continuous-time processes and can only approximate discrete probability mass function (PMFs) by using sufficiently fine binning.

However, we point out that **exact Dirac/pmf inter-event times are rarely realistic for physical continuous-time systems**: biological, chemical, and social processes typically exhibit continuous variability, and discrete PMFs usually arise from coarse measurement or binning rather than true atomic timing.

2. As I understand it, one of the importance of this paper is to provide an efficient way to simulate a non-Markovian multi-agent system with lower computational cost than directly simulating the renewal process for each agent, while keeping a high accuracy with appropriate parameter choices. But when the distribution is discrete, the simulation based on renewal processes only need to determine agents' states at integral time points. Then an associated question, in this case, is that would the proposed approach require more sampling than the traditional approach and thus higher computational cost?

If the underlying model is truly time-discrete, then a discrete-time renewal or DelaySSA implementation is indeed more efficient, because updates occur only at integer times and no continuous-time sampling is needed. MOSAIC is designed for **continuous-time, state-dependent, non-Markovian multi-agent systems**, where simulating each agent's renewal

process directly is expensive and MOSAIC provides substantial gains. For a strictly discrete-time model, however, MOSAIC would introduce unnecessary fine-scale sampling and is therefore **not the optimal choice**.

In response to point 1 and 2 of the reviewer, we added in the discussion:

“A further limitation arises when inter-event time distributions become highly peaked or nearly discrete (e.g., concentrated on multiples of a fixed time step): in this regime, representing the dynamics via a continuous-time hazard requires very large upper bounds in the thinning procedure, which leads to heavy rejection and poor efficiency. For strictly time-discrete models with atomic event times, dedicated discrete-time or DelaySSA-style algorithms remain better suited, and MOSAIC should be viewed only as a continuous-time approximation rather than the method of choice.”

3. I don't think the author correctly implements the spanning-tree method for constructing temporal networks. The key advance of this method is its ability to simultaneously reproduce burstiness in node and edge activities, and the corresponding exponents for nodes and the edge are adjustable according to the assigned activity pattern, which is exactly opposed to the statement in line 242.

We thank the referee for this clarification. The apparent discrepancy stems from a confusion between **edge interdurations** (the time between two consecutive activations of the same edge) and **edge durations** (the length of a single interaction once it is active). The spanning-tree method is indeed designed to reproduce burstiness and power-law behavior in node and edge interdurations, and the corresponding exponents can be tuned via the assigned activity patterns. This simultaneous and tunable control of node- and edge-level burstiness is in fact the main contribution of the spanning-tree construction, and was difficult to achieve in earlier temporal-network models; our original wording around line 242 did not adequately reflect this point.

In our analysis, however, we focused on *edge durations*, which in the spanning-tree construction are typically governed by a base rule (e.g. fixed or exponential) and therefore do not inherit the same heavy-tailed statistics as the interdurations. Thus, while spanning-tree correctly generates bursty edge interdurations, it does not, by default, reproduce non-trivial distributions of durations of individual interactions. MOSAIC-TN shares with spanning-tree a mathematical coupling between node and edge interdurations, because node and edge interdurations are analytically related (Supplementary Section C), so imposing bursty node inter-event distributions automatically yields bursty edge interdurations. Where MOSAIC-TN goes beyond spanning-tree is that it introduces a *separate channel* to control interaction

durations, allowing us to fit realistic, potentially non-exponential edge-duration distributions **independently** of the interduration statistics.

To clarify these points, we have revised Figure 3C to show the distribution of *edge interduations*. The updated figure confirms that both the spanning-tree method and MOSAIC-TN correctly reproduce the expected power-law behavior for edge interduations. At the same time, spanning-tree does not match the empirical distribution of edge durations, whereas MOSAIC-TN, via its dedicated duration channel, is able to reproduce both the observed interduration statistics and the distribution of interaction durations

We also added the following to the result section:

“In our pairwise instantaneous-rate framework, node and edge interduations are mathematically related (Supplementary Section C), and MOSAIC-TN reproduces the empirically observed burstiness at both levels. Previous models that incorporated bursty node IEDs treated interactions independently: nodes could display bursty activation while edge IEDs remained effectively memoryless. The spanning-tree framework captures edge-level burstiness but at the cost of reduced flexibility. MOSAIC-TN unifies these advantages by enforcing bursty node IEDs, reproducing bursty edge interduations, and retaining flexible, realistic interaction durations within a single coherent framework.”

4. There are several limitations in using the proposed method to generate temporal networks. For instance, first, the authors assume each pair of agents has a chance to interact at some point, but many real-world temporal networks have an underlying topology that constrains physical interactions between individuals, such as online social networks or transportation

networks. Some individuals can never interact because they are not connected within the underlying topology.

Second, the difficulty of generating temporal networks matched with empirical observations is the strong correlations between node and edge activations imposed by these underlying topologies. Mathematically, such a system should be modeled as a joint renewal process involving the activities of all nodes and edges, which cannot be decoupled into independent processes for each pair of nodes. But the proposed approach, at each time, simply samples two random individuals and determines their activations, ignoring the fact that the activity of these two nodes and the edge is affected by the other nodes and edges. In summary, I am fine with using the construction of temporal networks as an example, but there are clearly several limitations that the authors should acknowledge, for instance, this method only works for well-mixed underlying topologies.

We agree with the reviewer that our illustrative examples assume a fully mixed setting in which any pair of agents may interact. This choice was made solely to keep the demonstrations simple and transparent, but we acknowledge that it may be an oversimplification of real-world systems. This is also why we chose conference data for our main illustration: such events typically constitute relatively “open” social environments in which many participants can, in practice, interact with many others, making the fully mixed assumption a reasonable first approximation.

However, the MOSAIC-TN framework is fully compatible with arbitrary underlying topologies and non-independent node relationships: admissible interactions can be restricted to any predefined static network (social, spatial, infrastructural, etc.), and **the pair-selection step (step (iii) in MOSAIC-TN) is fully customizable**. In particular, one can (i) restrict the set of candidate pairs to those allowed by the desired topology and (ii) assign non-uniform probabilities when choosing the second node conditional on the first, thereby **encoding complex, non-independent relationships between nodes** or edges, as we do in MOSAIC-TN-B to model partner preference and history-dependent biases. Moreover, (iii) such relationships can also be encoded directly by scaling the pairwise rate $\Lambda(i,j)$ with a global network-topology factor $H(i,j)$, which can be updated at each iteration and can, for example, incorporate higher-order structure (e.g., second- or third-order relationships), while remaining flexible enough to capture arbitrary node- and edge-level attributes and memory effects.

In response to this comment, **we have revised the presentation of MOSAIC-TN** in the main text and Methods, and we have added a dedicated mathematical description of MOSAIC-TN with constrained topologies in the new Supplementary Section D.5. This section makes explicit how the rejection-based Gillespie scheme extends to arbitrary edge sets and non-

uniform pairing rules, while preserving the guarantees on node-level inter-event distributions.

Finally, **we have clarified in the Discussion** that the fully connected topology was chosen here purely for illustration, and that realistic applications should consider and exploit constrained topologies. We now explicitly note that incorporating such topologies is straightforward within MOSAIC-TN and can substantially improve computational efficiency, especially in large-scale HPC simulations (e.g., frameworks in the spirit of EpiFast or EpiHiper) where interactions are naturally restricted to sparse contact networks. Further exploration and empirical validation of constrained topologies is an important direction for future work.

“In our temporal-network examples, we focused on a fully connected topology and networks of up to 10^3 agents, primarily to keep the exposition and comparison with SSA-based baselines as simple and transparent as possible. This choice is made for illustration rather than realism, and many applications will naturally involve constrained, sparse, but large contact networks. Within MOSAIC-TN, such constrained topologies can be incorporated straightforwardly: candidate interactions can be restricted to edges of an arbitrary static network, and both the pair-selection mechanism and the pairwise kernel $\Lambda_{ij}(t_i, t_j)$ can encode dynamic topological structure. In that setting, the number of processes would scale with the number of admissible edges rather than the square of the population size, opening the door to larger systems on sparse graphs. Nonetheless, our current implementation is not an HPC framework and is not engineered for national-scale, distributed-memory simulations. Large-scale tools such as EpiFast and EpiHiper target a different regime, namely discrete-time dynamics on massive sparse networks. Systematically exploring sparse and application-specific topologies, and developing optimized implementations for that regime, is therefore an important direction for future work.”

Reviewer 2:

Reviewer 3:

The authors developed a stochastic simulation approach called MOSAIC (the abbreviation for modeling of stochastic agents with individual complexity), which can unify non-Markovian dynamics and agent heterogeneity. This approach extended the conventional Gillespie framework to non-Markovian systems, and also generalized the standard Gillespie algorithm by embedding heterogeneity directly at the level of individual agents. I think that the key of the approach is that the authors introduced the rejection probabilities of the involved events to reduce the computational complexity of non-Markovian systems. In addition, by only tracking a global maximum rate, the MOSAIC can efficiently scale to large populations and the interactions between multiple agents, while capturing diverse inter-event waiting-time distributions and individual-specific properties. Therefore, I think that the MOSAIC can be a general tool for stochastic simulation of various possible Markovian or non-Markovian systems across multidisciplinary. I appreciate the idea of this study and check all the mathematical equations, finding that they are correct. This manuscript is well written and organized. In a word, I would like to recommend it for acceptance. However, I have the following concerns and comments:

Three concerns

1. I think that in the MOSAIC, “N” is a key quantity to be determined since it directly influences the convergence of the approach. What the “N” represents seems unclear: It seems to me that it represents the number of “reaction channel” in some places, e.g., on line 312, but “the product of the largest possible molecular numbers of reaction species in other places, e.g., $N = N_A \times N_B$ ” on lines 55-56. If it is the former, then “N goes to infinitely” does not hold in some cases, e.g., gene-expression systems in general involves few reactions, and in this case, Δt would not tend to zero unless λ_{\max} is chosen to be very large. I suggest that the authors clearly describe how the “N” is determined and how the λ_{\max} is chosen.

We thank the reviewer for raising this point. In MOSAIC, N is not a tunable parameter. It is entirely determined by the system and corresponds to the number of elementary stochastic processes — that is, **the number of possible interactions or reaction channels in the model**. For multi-agent systems this is the number of admissible agent-agent interactions, and in chemical kinetics it corresponds to the number of ways reactants can be selected (for example, $N = N_A \times N_B$ for a bimolecular reaction). Thus, N simply reflects the combinatorial

size of the system and cannot be chosen independently. As the reviewer points out, N is never infinite and thus MOSAIC is not mathematically exact. However, the approximation error can be made arbitrarily small by increasing λ_{\max} , which, contrary to N , is freely chosen. We note that in MOSAIC, the error decreases very quickly as λ_{\max} increases (it scales like 1 divided by λ_{\max} squared), as visualized in Figure 2B.

Regarding the choice of λ_{\max} , in most of our examples the size of the systems under consideration naturally implied a large N . As a result, simply setting λ_{\max} to the actual maximum rate in the system was sufficient for the approximation error to be negligible. Only in Application 2, where N was around or below 30, did we explore increasing λ_{\max} . For systems with N larger than about 30, the error was already below numerical detectability (Figure 2B).

In addition, the rate λ_i (on line 71) does not depend on the system's state?

The rates λ_i do depend on the system state; in fact, they are what *define* the system state in MOSAIC. Each process can take any rate value at any iteration, and λ_i is simply the mechanism used to encode these state-dependent rates. We have clarified this point in the revised section 1.1 of manuscript, when we introduce MOSAIC.

2. The algorithmic procedure described on lines 70-77 is difficult to understand (but I know that this procedure only described the algorithm that the time advances only by Δt) (one single time step). What is the relationship between the elapsed time t_j since the last event of the j th process and the time increment Δt ? The authors did not clearly show how the time advances to arrive at the moment t_j by increasing the time increment Δt . Is the elapsed time t_j since the last event of the j th process known? I think that clarifying this point is very important for understanding the MOSAIC.

We thank the reviewer for pointing out that the description of the algorithmic steps may have been unclear. MOSAIC explicitly tracks the elapsed time t_j since the last event of process j .

Namely, at each iteration, when the global time advances by an increment Δt (sampled from the exponential distribution defined by λ_{\max}), every process j updates its elapsed time by $t_j \rightarrow t_j + \Delta t$, except for the single process that fires, whose elapsed time is reset to zero. Thus, t_j is simply the running total of all Δt increments since the last event of that process; time does not “jump” directly to t_j .

We note that because MOSAIC tracks the elapsed time for every process, the space complexity is $O(N)$. We have revised section 1.1 of manuscript to state explicitly that these elapsed times are tracked and to clarify the time-update mechanism.

3. In the case that the “ N ” tends to infinity, the number of the loops in the algorithmic procedure would be very large. Thus, the MOSAIC would not be applied in the case of very large “ N ”. Is my understanding correct?

The reviewer is correct. MOSAIC, like any stochastic simulation algorithm that simulates events one by one, is not appropriate for extremely large systems. The expected number of rejection loops per accepted event does not grow with N , but the overall computational cost per unit simulated time still scales linearly with N because we are tracking and updating N processes. In other words, for very large N the number of time steps required to reach a given final time becomes very large, which is a generic limitation of event-by-event SSA / Gillespie style methods rather than a MOSAIC-specific issue.

For such large regimes, piecewise-deterministic Markov process (PDMP) approaches and related hybrid methods often provide a more suitable alternative: they simulate rare events stochastically while treating frequent or high-intensity dynamics deterministically, leading to substantial speedups in very large systems.

We have extended the Discussion to address this:

“More generally, all stochastic simulation frameworks face challenges when applied to systems with very large numbers of reactants and reaction channels. In such cases, analytical simplifications or surrogate approaches, such as training neural networks on limited simulations to generate large-scale ensembles, can reduce computational cost. Another fundamental difficulty arises when reaction propensities span several orders of magnitude: the dominance of high-frequency events can make rare events prohibitively expensive to observe, as noted in RNA transcription studies. Piecewise-deterministic Markov process (PDMP) models provide an effective alternative by simulating rare events stochastically while handling frequent events deterministically, leading to major speedups in diverse applications. Hybrid extensions of MOSAIC that combine these ideas represent a natural next step.”

In addition, it seems to me that the MOSAIC does not consider the reaction system's state, unlike the standard Gillespie algorithm which explicitly considers the system's state. I cannot understand this point. Can you clarify it or give some interpretations?

We would like to clarify that MOSAIC does indeed consider the reaction system's state, analogously to the standard Gillespie algorithm. As in the standard SSA, where propensities depend on the state vector, MOSAIC's rates λ_i are allowed to depend arbitrarily on the current state and on the elapsed times t_i . In our formulation, the state is encoded through these $\lambda_i(t, \text{state})$ and through the specified state-update rule when a given process fires. We have clarified this state dependence more explicitly in the revised section 1.1 of the manuscript.

In light of reviewers' remarks 1–3, **we have substantially revised the section where we introduce MOSAIC (Section 1.1)**. In particular, we now (i) clearly define (N) as the number of elementary stochastic processes determined by the model structure and emphasize that it is not a tunable parameter; (ii) clarify the role of λ_{\max} as a user-chosen global upper bound that controls the accuracy–efficiency trade-off; (iii) make explicit how the elapsed times t_j are updated via a global clock and last-event times; and (iv) state more precisely how state dependence of the rates λ_j is handled without maintaining the full vector of propensities at each step.

Minor comments:

1. Applications I and II, although representative, are actually small systems. For each of these systems, how is the λ_{\max} chosen and what size?

We thank the reviewer for this question. The system sizes and choices of λ_{\max} for Applications I and II are as follows.

In Application 1 (affinity maturation), we benchmarked across different system sizes, as shown in Figure 1D. A representative regime is $N_B = 1000$ B cells and $N_T = 10$ T cells, giving $N \approx 10,000$ pair-specific processes. In this example the dynamics are purely Markovian, so we simply set λ_{\max} equal to the true maximum rate over all processes. In this regime MOSAIC is exact, and there is no benefit in artificially increasing λ_{\max} . **We have added a short sentence in Section 1.2 to make this choice explicit.**

“In this application the dynamics are purely Markovian, so the simulation is exact and we simply set $\lambda_{\max} = \max_{1 \leq j \leq N} \lambda_j$, i.e., equal to the true maximum rate over all processes.”

In Application 2 (gene expression with delays), the effective number of processes is smaller (typically on the order of $N \approx 100$ when summing mRNA and protein processes), and this is precisely where different choices of λ_{\max} are explored. We systematically varied λ_{\max} and reported how this affects both accuracy and runtime (Figure 2B), illustrating how λ_{\max} can be used to tune the approximation error in smaller systems. **Given the importance of this parameter, we have expanded our discussion of λ_{\max} for application 2 in the revised manuscript** (both the results and method section).

“Stricter choices of λ_{\max} further reduce errors in small systems, at only modest additional computational cost (Supplementary Section B.4); we find here that increasing λ_{\max} above the true maximum rate is only beneficial for systems with fewer than about 20 processes.”

And in the method section:

For the mRNA elongation reaction, the waiting time τ follows a Gamma distribution $\mathcal{G}(\lambda_0, \alpha_{\text{elong}})$ which we model in MOSAIC with the instantaneous rate

$$\lambda_{\text{elong}}(t) = \frac{\psi_{\text{elong}}(t)}{1 - F_{\text{elong}}(t)} = \frac{(\alpha_{\text{elong}} \lambda_0)^{\alpha_{\text{elong}}} t^{\alpha_{\text{elong}} - 1} e^{-\alpha_{\text{elong}} \lambda_0 t}}{\Gamma(\alpha_{\text{elong}}, \alpha_{\text{elong}} \lambda_0 t)} \leq \lambda_{\max}^{(\text{elong})} = \alpha_{\text{elong}} \lambda_0, \quad [10]$$

where $\Gamma(\cdot, \cdot)$ denotes the upper incomplete gamma function, and $\lambda_{\max}^{(\text{elong})}$ the global upper bound for the elongation channel in MOSAIC. For small systems, where $N = N_N + N_M + N_P$ is modest, we also evaluated a more conservative choice to ensure a smaller effective time step Δt (Supplementary Section B.4), by enforcing

$$N \lambda_{\max}^{(\text{elong})} \geq 20 \lambda_0, \quad \text{which implies an average time step } \mathbb{E}[\Delta t] \leq \frac{1}{20 \lambda_0}. \quad [11]$$

2. On lines 136-137, the description “In the standard Gillespie algorithm, this requires tracking $N = N_B \cdot N_T$ reaction channels” seems incorrect. Regarding “reaction channel”, the common understanding is that one reaction corresponds to one reaction channel. Application III is a “good” example since it can better interpret the implement of MOSAIC, and I think that this example is interesting.

We agree that in the traditional formulation of the Gillespie algorithm, where all molecules of a given species are statistically identical, it is indeed standard to speak of one “reaction channel” per reaction. In our affinity-maturation model, however, individual B cells are **not equivalent**: each B cell has its own affinity, and the effective rate of interaction depends on the specific B–T pair. From an SSA perspective, each admissible B–T pair is therefore a distinct stochastic process with its own propensity, and we refer to these pair-specific

processes as “reaction channels,” giving $N = N_B \times N_T$. **We have clarified this sentence in the revised manuscript to avoid confusion as follow:**

“In a standard Gillespie-style implementation for this heterogeneous setting, one must treat each admissible B–T pair as a separate stochastic process with its own propensity, leading to $N = N_B \times N_T$ such processes and $O(N)$ complexity per step.”

3. For a general biochemical reaction network, the authors [JJ Zhang, TS Zhou. Markovian approaches modeling intracellular reaction processes with molecular memory. PNAS 116, 47:23542-23550(2019)] used general waiting-time distributions to model the occurrence of inter-events. I suggest that the authors of this manuscript cite that paper.

We thank the reviewer for this helpful suggestion. Zhang and Zhou (PNAS, 2019) provide an important analytical framework for studying reaction networks with general waiting-time distributions. This work is highly relevant to our discussion, and **we have now cited this paper** in both the Introduction and the Discussion of the revised manuscript.

4. The abstract seems a little long, and I suggest that the authors shorten it so that it is not more than 200 words.

We have now shortened the abstract to less than 200 words.

Reviewer 4:

Summary: The paper describes MOSAIC (Modeling of Stochastic Agents with Individual Complexity). MOSAIC is a general and scalable framework that embeds agent-specific properties directly into the dynamics. MOSAIC has several key distinguishing features – it unifies heterogeneous rates, interaction preferences, and diverse waiting-time distributions within a single stochastic formalism. The new capabilities enable multiscale modeling in which microscopic variability drives emergent macroscopic behavior, a capability currently inaccessible to standard simulation algorithms. The authors claim that MOSAIC can be combined with other known methods such as Gillespie algorithm to develop efficient large-scale simulations. The authors describe applications to biochemical reactions, immune cell dynamics, and temporal social networks to show that MOSAIC efficiently reproduces empirical features that existing methods either miss or capture only at prohibitive computational cost, establishing it as a robust tool for the next generation of stochastic simulations.

Overall evaluation: The approach is interesting and well founded (as described in the Appendix). The primary contribution of the paper appears to be theoretical in nature however and the authors have provided proof theoretic results; this is a very strong aspect of the paper. However, the stated efficiency gains are not clear as discussed. Specifically, a detailed computational comparison is missing. The flexibility of the approach is noteworthy, but other approaches dating back several years have reported on similar ideas.

The main idea is simple and powerful and is described nicely in the paper: the MOSAIC algorithm chooses a single global maximum rate, samples candidates process uniformly at random and then accepted at their instantaneous rate. Proofs in the Appendix show that this approach overcomes the known challenges with earlier methods. The paper needs experimental results to illustrate the gains one can hope to get in practice and as claimed. Comparisons with systems that have been already deployed should be discussed to further illustrate the significance.

We thank the reviewer for the positive assessment of the theoretical contribution and for highlighting the importance of experimental validation. Although the core of the paper is theoretical, we do include extensive simulation experiments in each application section. In particular, each use case is calibrated to experimentally or empirically grounded systems, and we benchmark MOSAIC against two or three of the most widely used method families for that domain (e.g., next-reaction methods, delay-based algorithms, and ABMs). These experiments demonstrate the practical speedups and accuracy gains of MOSAIC in settings that are representative of real multi-agent and biochemical systems. In the revised manuscript, we have expanded the discussion to more clearly highlight these experimental results and relate them to existing deployed frameworks in those domains.

“We have demonstrated MOSAIC’s versatility across three distinct domains. First, in a system of B cell competition and evolution, MOSAIC efficiently models clonal selection driven by B cell receptor diversity. [...] Second, in an RNA transcription system with delayed negative feedback, MOSAIC captures state-dependent, non-exponential waiting times that classical delay-based algorithms cannot dynamically adjust. [...] Third, in temporal social networks, MOSAIC-TN captures node-driven activity patterns in which individuals alternate between long inactive periods and short bursts of interactions. [...] Together, these use cases show that MOSAIC unifies heterogeneity and memory in a single scalable framework, recovering empirical features that existing methods approximate poorly or fail to capture altogether.”

Second is the value of global maximum rate. This rate shows up in the proofs (as it should) and seems like a necessary parameter. Can the authors say a bit more about the dependency.

We thank the reviewer for raising this important point. As we discuss in our response to Reviewer 3, the global maximum rate (λ_{\max}) is the only parameter that can be explicitly adjusted to tune the accuracy of MOSAIC. For correctness, λ_{\max} must be at least the true maximum instantaneous rate across all processes. Increasing λ_{\max} beyond this lower bound monotonically improves the approximation of the target non-Markovian dynamics (at the cost of a higher rejection rate), and we provide an explicit error bound as a function of λ_{\max} in the Supplementary Information (section B.4).

In most practical systems, the setting λ_{\max} to its lower bound is already sufficient to yield highly accurate simulations, because the number of processes is sufficiently large to make the error insignificant (<1%). In our toy example (Supplementary section B.4), we found that $N \geq 30$ yields < 1 % error in the inter-event-time distribution (IED).

However, in small systems, where the number of processes N is moderate, explicitly increasing λ_{\max} can be necessary for reducing the approximation error. Hence, λ_{\max} provides a direct and transparent accuracy–efficiency trade-off: higher λ_{\max} decreases the simulation error but increases the expected number of candidates draws.

The reviewer makes a good observation that the magnitude of λ_{\max} depends strongly on the chosen IED. Heavy-tailed distributions typically lead to modest rates, whereas distributions with sharp, spike-like behavior can produce sudden spike of very large instantaneous rates, which in turn affects the rejection rate in the thinning step. To make this dependency clearer, we have added a discussion in the revised manuscript explaining how the choice of the IED influences λ_{\max} and what this implies for efficiency.

“A further limitation arises when inter-event time distributions become highly peaked or nearly discrete (e.g., concentrated on multiples of a fixed time step): in this regime, representing the dynamics via a continuous-time hazard requires very large upper bounds in the thinning procedure, which leads to heavy rejection and poor efficiency. For strictly time-discrete models with atomic event times, dedicated discrete-time or DelaySSA-style algorithms remain better suited, and MOSAIC should be viewed only as a continuous-time approximation rather than the method of choice.”

Detailed comments:

- In the abstract: “where inter-event times follow exponential distributions” The Gillespie algorithm is shown to be correct on non-mass-action kinetics: Sanft KR, Gillespie DT, Petzold LR. Legitimacy of the stochastic Michaelis-Menten approximation. IET Syst Biol. 2011 Jan;5(1):58. doi: 10.1049/iet-syb.2009.0057. PMID: 21261403. In fact, it is correct for all kinetics which are not time dependent. This fact has been utilized in the Systems Biology community since 2006.
- Lines: 8-9: As stated in the abstract the Gillespie algorithm is applicable for non-exponential waiting time. This discrepancy appears several times in the paper and needs to be either fixed or further clarified.

We thank the reviewer for pointing out the source of confusion. Our intent was to highlight the limitation of the classical Gillespie SSA for non-Markovian processes, not to imply that it applies only to mass-action kinetics. We fully agree that the SSA is exact for any reaction network whose propensities depend only on the current state and not on the history or elapsed time, including systems with non-mass-action kinetics such as Michaelis-Menten or Hill-type rates, as shown for example in Sanft et al. (2011). In this Markovian setting, the waiting time to the next reaction is exponentially distributed *conditional on the instantaneous state*; apparent non-exponential “kinetics” at the level of aggregate statistics arise solely because the system visits different states with different total propensities. In contrast, in the non-Markovian systems we consider, the event hazard depends explicitly on the age of a process or on its history, so that the waiting time is no longer exponential even when conditioned on the current state.

To avoid misunderstanding, we have revised the manuscript to state clearly that the limitation of the classical SSA concerns non-Markovian dynamics, i.e., systems in which the propensities cannot be written purely as functions of the instantaneous state of the system. We have removed unnecessary mentions of the exponential distribution from the manuscript to prevent confusion.

In the abstract:

“Stochastic processes underpin dynamics across biology, physics, epidemiology, and finance, yet accurately simulating them remains a major challenge. Classical approaches such as the Gillespie algorithm are exact for Markovian, time-independent systems, where propensities depend only on the current state and agents of a given type are statistically identical.”

In the introduction:

“Yet, the simplicity of the Markovian paradigm comes at a cost. Real systems rarely behave in a memoryless way, and most cannot be reduced to propensities that depend only on the instantaneous state.”

- Lines: 1-2: Chemistry should be mentioned especially since Gillespie algorithm is equivalent to the chemical master equation.

Thanks for the comment. We have added chemistry to the text.

- Lines: 27-28: EpiHiper should be mentioned in this context: Chen J, Hoops S, Mortveit HS, Lewis BL, Machi D, Bhattacharya P, Venkatramanan S, Wilson ML, Barrett CL, Marathe MV. EpiHiper-A high performance computational modeling framework to support epidemic science. PNAS Nexus. 2024 Dec 11;4(1):pgae557. doi: 10.1093/pnasnexus/pgae557. PMID: 39720202; PMCID: PMC11667244.

Thanks for the reference. We now mention EpiHiper in the text.

- Lines: 32-34: Prohibitive computational cost is unclear, especially since there is no comparison provided.

We thank the reviewer for pointing out that the phrase “prohibitive computational cost” is too strong without a direct quantitative comparison. Our intent was not to suggest that existing ABM or HPC frameworks are universally impractical, but rather that they typically require large event queues, complex parallelization, and substantial hardware resources when used to model heterogeneous, agent-level dynamics, at scale. In contrast, MOSAIC targets a different regime: it provides agent-level heterogeneity and non-Markovian timing within a single-node, rejection-based SSA framework, with per-event cost comparable to classical Gillespie methods.

To avoid overstatements, we have softened the wording in the manuscript and now emphasize the **algorithmic simplicity and single-node scalability** of MOSAIC rather than characterizing other approaches as prohibitive.

- 103-147: A performance comparison with a tool like COPASI would be necessary to support the claim of efficiency. Stefan Hoops, Sven Sahle, Ralph Gauges, Christine Lee, Jürgen Pahle, Natalia Simus, Mudita Singhal, Liang Xu, Pedro Mendes, Ursula Kummer, COPASI—a COmplex PATHway Simulator, Bioinformatics, Volume 22, Issue 24, December 2006, Pages 3067–3074, <https://doi.org/10.1093/bioinformatics/btl485>

- Line: 152: A tool like COPASI would have no problem simulating these types of models. The argument of the extension to non-exponential IEDs is not convincing.

The affinity-maturation model in Application 1 is an **agent-based competitive system with explicit individual-level heterogeneity**: each B cell carries its own affinity value, these affinities evolve over time, and competition depends on pairwise comparisons of affinities (a B cell can displace another only if its affinity is strictly higher). Expressing such a system in COPASI would require introducing one species per individual B cell and one reaction for every ordered pair of potentially interacting cells. For $N_B = 1000$ B cells and $N_T = 10$ T cells, this translates into **tens of thousands** of interactions, many with state-dependent conditional rules. In practice, representing these conditional interactions in COPASI would require introducing a very large number of reactions, and its solvers are neither optimized nor efficient for models with such extensive combinatorial heterogeneity.

In contrast, MOSAIC handles these interactions directly at the **agent level**, with individual properties, state-dependent rates, and event-rejection rules that naturally capture the “affinity > existing affinity” condition without enumerating all possible pairs. As a result, MOSAIC can efficiently simulate the full $N_B \times N_T$ competitive landscape, whereas COPASI would require explicit encoding of every reaction channel. For example, a system with 10,000 B cells and 100 T cells would translate to roughly one million distinct reaction channels in COPASI, making this approach impractical at scale.

Regarding non-exponential inter-event times: we agree that COPASI supports fixed and simple delay mechanisms, but it does not support **general age-dependent hazards or arbitrary non-exponential waiting-time distributions**. These forms of molecular memory are central to later sections of the manuscript, and MOSAIC can handle them natively. COPASI cannot do so without major approximations or manual workarounds.

For these reasons, COPASI is not well-suited as a baseline for the agent-level heterogeneous model used in Application 1, and a performance comparison would not be informative. We have added clarification to the section 1.2 of the manuscript to make this distinction explicit.

“Next, we examine computational scaling. Conventional Gillespie methods and its variant like COPASI face a combinatorial burden due to pairwise competition. After each event, it is necessary to check whether any unbound B cell has higher affinity than those currently bound to T cells, which requires maintaining and updating a list of competing pairs. Because each displacement event can change the rates of many potential competitors, this bookkeeping becomes computationally expensive. In a standard Gillespie-style implementation for this heterogeneous setting, one must treat each admissible B–T pair as a

separate stochastic process with its own propensity, leading to $NB \times NT$ such processes and $O(N)$ complexity per step.”

- Lines: 194-225: The number of nodes is limited to $1e3$ which not very convincing when compared to tools like EpiFast or EpiHiper that deal with millions of nodes though. Again, a comparison is necessary before claiming efficiency.

We thank the reviewer for this comment. The goal of the epidemic example in our manuscript is not to demonstrate national-scale HPC performance—unlike EpiFast or EpiHiper, which are specialized, discrete-time frameworks engineered to simulate millions of agents—but rather to illustrate that MOSAIC-TN can efficiently generate continuous-time, non-Markovian temporal networks with arbitrary inter-event-time distributions. These capabilities are fundamentally different from those provided by EpiFast/EpiHiper, which rely on discrete-time updates and typically assume Markovian or discretely semi-Markov transitions.

In its current form, MOSAIC-TN is not designed for national-scale networks, because of the number of processes it must track scales with the number of possible interactions between agents. In a fully connected or well-mixed network, this number grows quadratically with the number of agents, making a million-node network computationally infeasible for any event-by-event SSA-based method. In principle, MOSAIC-TN can also operate on sparse underlying topologies by restricting the admissible edges, in which case the number of processes would scale with the number of edges, as in large-scale ABMs. However, topology-constrained temporal networks and HPC implementations are beyond the scope of the present work, and our current implementation is not optimized for national-scale distributed-memory simulations.

For these reasons, we do not consider frameworks such as EpiFast and EpiHiper to be appropriate baselines for our study: they address a fundamentally different computational problem (large-scale, discrete-time epidemic ABM on sparse or structured networks), whereas MOSAIC-TN targets accurate continuous-time non-Markovian dynamics at small to moderate scales. Our experiments therefore focus on regimes where continuous-time non-Markovian fidelity is essential and where comparisons with other SSA-based methods are appropriate. We have clarified this distinction and **explicitly stated this scalability limitation in the Discussion of the revised manuscript.**

“In our temporal-network examples, we focused on a fully connected topology and networks of up to 10^3 agents, primarily to keep the exposition and comparison with SSA-based baselines as simple and transparent as possible. This choice is made for illustration rather than realism, and many applications will naturally involve constrained, sparse, but large

contact networks. Within MOSAIC-TN, such constrained topologies can be incorporated straightforwardly: candidate interactions can be restricted to edges of an arbitrary static network, and both the pair-selection mechanism and the pairwise kernel $\Lambda_{ij}(t_i, t_j)$ can encode dynamic topological structure. In that setting, the number of processes would scale with the number of admissible edges rather than the square of the population size, opening the door to larger systems on sparse graphs. Nonetheless, our current implementation is not an HPC framework and is not engineered for national-scale, distributed-memory simulations. Large-scale tools such as EpiFast and EpiHiper target a different regime, namely discrete-time dynamics on massive sparse networks. Systematically exploring sparse and application-specific topologies, and developing optimized implementations for that regime, is therefore an important direction for future work.”

Unifying non-Markovian Dynamics and Agent Heterogeneity in Scalable Stochastic Simulation

A. Pelissier, M. Phan, D. Le Bail, N. Beerenwinkel, R. Martinez.

Summary: The paper describes MOSAIC (Modeling of Stochastic Agents with Individual Complexity). MOSAIC is a general and scalable framework that embeds agent-specific properties directly into the dynamics. MOSAIC has several key distinguishing features – it unifies heterogeneous rates, interaction preferences, and diverse waiting-time distributions within a single stochastic formalism. The new capabilities enable multiscale modeling in which microscopic variability drives emergent macroscopic behavior, a capability currently inaccessible to standard simulation algorithms. The authors claim that MOSAIC can be combined with other known methods such as Gillespie algorithm to develop efficient large-scale simulations. The authors describe applications to biochemical reactions, immune cell dynamics, and temporal social networks to show that MOSAIC efficiently reproduces empirical features that existing methods either miss or capture only at prohibitive computational cost, establishing it as a robust tool for the next generation of stochastic simulations.

Overall evaluation: The approach is interesting and well founded (as described in the Appendix). The primary contribution of the paper appears to be theoretical in nature however and the authors have provided proof theoretic results; this is a very strong aspect of the paper. However, the stated efficiency gains are not clear as discussed. Specifically, a detailed computational comparison is missing. The flexibility of the approach is noteworthy, but other approaches dating back several years have reported on similar ideas.

The main idea is simple and powerful and is described nicely in the paper: the MOSAIC algorithm chooses a single global maximum rate, samples candidates processes uniformly at random and then accepted at their instantaneous rate. Proofs in the Appendix show that this approach overcomes the known challenges with earlier methods.

The paper needs experimental results to illustrate the gains one can hope to get in practice and as claimed. Comparisons with systems that have been already deployed should be discussed to further illustrate the significance. Second is the value of global maximum rate. This rate shows up in the proofs (as it should) and seems like a necessary parameter. Can the authors say a bit more about the dependency.

Detailed comments

- In the abstract: “where inter-event times follow exponential distributions” The Gillespie algorithm is shown to be correct non-mass-action kinetics: Sanft KR, Gillespie DT, Petzold LR. Legitimacy of the stochastic Michaelis-Menten approximation. *IET Syst Biol.* 2011 Jan;5(1):58. doi: 10.1049/iet-syb.2009.0057. PMID: 21261403. In fact, it is correct for all kinetics which are not time dependent. This fact has been utilized in the Systems Biology community since 2006.
- Lines: 1-2: Chemistry should be mentioned especially since Gillespie algorithm is equivalent to the chemical master equation.
- Lines: 8-9: As stated in the abstract the Gillespie algorithm is applicable for non-exponential waiting time. This discrepancy appears several times in the paper and needs to be either fixed or further clarified.
- Lines: 27-28: EpiHiper should be mentioned in this context: Chen J, Hoops S, Mortveit HS, Lewis BL, Machi D, Bhattacharya P, Venkatramanan S, Wilson ML, Barrett CL, Marathe MV. EpiHiper-A high performance computational modeling framework to support epidemic science. *PNAS Nexus.* 2024 Dec 11;4(1):pgae557. doi: 10.1093/pnasnexus/pgae557. PMID: 39720202; PMCID: PMC11667244.
- Lines: 32-34: Prohibitive computational cost is unclear, especially since there is no comparison provided.
- 103-147: A performance comparison with a tool like COPASI would be necessary to support the claim of efficiency. Stefan Hoops, Sven Sahle, Ralph Gauges, Christine Lee, Jürgen Pahle, Natalia Simus, Mudita Singhal, Liang Xu, Pedro Mendes, Ursula Kummer, COPASI—a COMplex PATHway SIMulator, *Bioinformatics*, Volume 22, Issue 24, December 2006, Pages 3067–3074, <https://doi.org/10.1093/bioinformatics/btl485>
- Line: 152: A tool like COPASI would have no problem simulating these types of models. The argument of the extension to non-exponential IEDs is not convincing.

- Lines: 194-225: The number of nodes is limited to $1e3$ which not very convincing when compared to tools like EpiFast or EpiHiper that deal with millions of nodes though. Again, a comparison is necessary before claiming efficiency.